# Comparison of ice dynamics using full-Stokes and Blatter-Pattyn approximation: application to the Northeast Greenland Ice Stream

Martin Rückamp[1,*], Thomas Kleiner[1], and Angelika Humbert[1,2]

[1]Alfred-Wegener-Institut Helmholtz-Zentrum für Polar- und Meeresforschung, Bremerhaven, Germany
[2]Faculty of Geoscience, University of Bremen, Bremen, Germany
[*]now at Bavarian Academy of Sciences and Humanities, Munich, Germany

**Correspondence:** Martin Rückamp (martin.rueckamp@badw.de), Angelika Humbert (angelika.humbert@awi.de)

**Abstract.** Full-Stokes (FS) ice sheet models provide the most sophisticated formulation of ice sheet flow. However, their applicability is often limited due to the high computational demand and numerical challenges. To balance computational demand and accuracy, the so-called Blatter-Pattyn (BP) stress regime is frequently used. Here, we explore the dynamic consequences of using simplified approaches by solving FS and the BP stress regime applied to the Northeast Greenland Ice Stream. To ensure a consistent comparison, we use one single ice sheet model to run the simulations under identical numerical conditions. A sensitivity study to the horizontal grid resolution (from 12.8 to a resolution of 0.1 km) reveals that velocity differences between the FS and BP solution emerge below ∼1 km horizontal resolution and continuously increase with resolution. Over the majority of the modelling domain both models reveal similar surface velocity patterns. At the grounding line of 79° North Glacier the simulations show considerable differences whereby the BP model overestimates ice discharge of up to 50% compared to FS. A sensitivity study to the friction type reveals that differences are stronger for a power-law friction than a linear friction law. Model differences are attributed to topographic variability and the basal drag, where neglected stress terms in BP become important.

## 1 Introduction

The most comprehensive description of ice sheet flow is given by the full-Stokes (FS) equations. Such a model is considered the most accurate available, capable of describing highly dynamic ice sheets, including ice streams, ice shelves, and grounding line migration. But this formulation is also the most computationally expensive, both from the numerical perspective and because FS effects only occur at higher resolutions.

Due to increasing computer power, the emergence of new generation ice sheet models (ISMs) in the last decades allowed for ice flow models to directly rely on FS equations or approaches with only few simplifications. Although ice dynamical processes require an accurate representation of the stresses, e.g. in regions of high variability in basal topography and/or basal slipperiness (Gudmundsson, 2003; Hindmarsh, 2004), simplifications to FS are still necessary long time integrations or for large ensemble

modelling to keep the computing time affordable. For this reason, ice flow modelling has frequently relied on simplified mathematical models to balance computational costs and accuracy, such as the three-dimensional and computationally efficient Blatter-Pattyn (BP) formulation (Blatter, 1995; Pattyn, 2003). Although BP neglects several components of the full system (e.g., bridging stresses), it is considered valid in the majority of an ice sheet.

Current ice sheet projections within the ISMIP6 effort (Ice Sheet Model Intercomparison Project for CMIP6 (ISMIP6), Nowicki et al., 2016) cover a large diversity in model approaches and approximations to the Stokes flow (Goelzer et al., 2018; Seroussi et al., 2019; Goelzer et al., 2020; Seroussi et al., 2020), but only one contribution solved the FS problem (Seroussi et al., 2020, model *UTAS_ElmerIce* with a variable resolution between 4 and 40 km). Up to now, there is no clear understanding of whether FS simulations have potential in narrowing uncertainties in current sea-level predictions from the Greenland and Antarctic ice sheets. However, the ability to accurately simulate the behaviour of outlet glaciers has been cited in the reports of the Intergovernmental Panel on Climate Change as a prerequisite for reliable ice sheet projections (IPCC, 2013; Meredith et al., 2019; Oppenheimer et al., 2019).

In earlier ISMIPs the performance in simulating idealized ice stream and marine ice sheet dynamics by ISMs was explored: e.g. in ISMIP-HOM (Ice Sheet Model Intercomparison Project for Higher-Order Models, Pattyn et al., 2008), MISMIP2d (Marine Ice Sheet Model Intercomparison Project, Pattyn et al., 2012), MISMIP3d (Pattyn et al., 2013), MISMIP+ (Cornford et al., 2020). The ISMIP-HOM exercise focused on so-called higher-order (HO) as well as FS model and in particularly reveals that the FS models show a smaller spread, hence are in better agreement with one another and with available analytical solution. In the earlier MISMIP experiments, any systematic study of differences in solutions from Stokes models was not possible due to the limited number of FS models[1]. The recent MISMIP+ effort reflects that the choice of the friction law plays a larger role compared to, e.g. ice flow approximations (two FS models contribute to this exercise). From these synthetic scenarios, it yet needs to be assessed if discrepancies between FS and simpler models are minor compared to other uncertainties in ice sheet modelling given the huge additional computational cost when running FS models (about 10 times slower than BP (Larour et al., 2012, Table 3 therein)).

In contrast to the idealized ISMIP experiments, the influence of FS on ice dynamical behaviour compared to simpler models was tested on realistic problems (e.g. Leysinger Vieli and Gudmundsson, 2004; Le Meur et al., 2004; Morlighem et al., 2010; Seddik et al., 2012; Favier et al., 2014; Seddik et al., 2017). These exercises provide important insights into the transient response and indicate that FS produces different results compared to simpler models. For example, in most of the conducted projection experiments, FS tends to contribute less to sea-level rise than simpler models under identical projection scenarios (Seddik et al., 2012; Favier et al., 2014; Seddik et al., 2017). Morlighem et al. (2010) concluded that treating ice flow of the fast-flowing Pine Island Glacier (PIG, Antarctica) with a steeply rising bed near the grounding line with FS is essential. However, assessing whether FS is urgently needed compared to simpler models is complicated because of many interacting processes: (i) comparability is often limited by the use of different ISMs as it is not entirely clear how much of the differences is due to numerical treatments as, e.g. different horizontal grids resolve the bed differently, or discretization schemes. (ii) Differences are also subject to e.g. assumptions in basal flow conditions or the initialization approach. (iii) Moreover, when

---

[1]Compared to the ISMIP-HOM experiments, the MISMIP experiments require a long transient integration, which is costly to perform for an FS model.

transient simulations are forced with climate scenarios, the response involves numerous processes and interactions that make it difficult to separate causes.

Here, we intend to overcome model-intercomparison shortcomings by performing consistent numerical experiments by solving the FS system or using the BP approximation. Our comparison focuses on FS and BP, as the BP is used as a compromise for FS to balance computational costs and accuracy. The employed ISM includes the option to run the FS model or to activate the BP mode allowing us to compare the results consistently as they are computed under the same numerical conditions. Here, we strive to fill the gap of FS and BP intercomparisons, stepping away from synthetic scenarios, and performing a high-resolution real system application. We select two regions of the Northeast Greenland Ice Stream (NEGIS) as an investigation area. Both regions cover slow and fast flowing ice as well as smooth and highly variable bed topography. The first domain focuses on the ice stream while the second domain covers the NEGIS outlets 79°N Glacier (79NG) and Zacharias Isbræ (ZI). Our study does not treat the transient behaviour, we rather aim to address ice dynamical differences between the stress regimes. As differences between FS and BP are expected to emerge in regions with steep bed gradients or high aspect ratios (height/length ratio $\epsilon$), we step-wise increase spatial resolution from 12.8 km to a resolution of 0.1 km.

## 2 Theoretical background

### 2.1 The full-Stokes equations

In the case of ice sheets, the Reynolds number is very low (e.g., Fowler and Larson, 1978) and therefore the inertial term in the Navier–Stokes equations can be neglected. This approximation is often referred to as Stokes flow. The core of the FS equation system builds the mass balance and momentum equation for incompressible ice and is written as

$$\operatorname{div} \boldsymbol{v} = 0, \tag{1}$$

$$\operatorname{div} \mathbf{t} = \varrho_i \, \boldsymbol{g}, \tag{2}$$

with the density of ice $\varrho_i = 917 \, \mathrm{kg \, m^{-3}}$, the three dimensional velocity field $\boldsymbol{v} = (v_x, v_y, v_z)$ in Cartesian coordinates, the gravitational acceleration vector pointing downward $\boldsymbol{g} = -g \boldsymbol{e}_z$ ($g = 9.81 \, \mathrm{m \, s^{-2}}$), and the Cauchy stress tensor $\boldsymbol{t}$. We split the Cauchy stress into a deviatoric part $\mathbf{t}^{\mathrm{D}}$ and an isotropic pressure $p$

$$\mathbf{t} = \mathbf{t}^{\mathrm{D}} + p \boldsymbol{I}, \tag{3}$$

with $p = -\frac{1}{3} \operatorname{tr}(\mathbf{t})$ and $\boldsymbol{I}$ the identity tensor. The constitutive equation for the non-Newtonian fluid links strain rates to velocity gradients

$$\mathbf{t}^{\mathrm{D}} = 2\eta \boldsymbol{D}, \tag{4}$$

where $\boldsymbol{D}$ is the strain rate tensor ($\boldsymbol{D} = \frac{1}{2}(\operatorname{grad} \boldsymbol{v} + (\operatorname{grad} \boldsymbol{v})^T)$). The viscosity is given by the Glen-Steinemanns flow law (Glen, 1955; Steinemann, 1954)

$$\eta = \frac{1}{2} E A(T, W)^{-1/n} \dot{\varepsilon}_e^{(1-n)/n}, \tag{5}$$

with the flow law exponent $n = 3$, the enhancement factor $E$, the rate factor $A$ depending on temperature $T$ and microscopic water content $W$, and the effective strain rate $\dot{\varepsilon}_e$ being the second invariant of the strain-rate tensor. To the latter, a small value of $\dot{\varepsilon}_0 = 10^{-30}\,\mathrm{s}^{-1}$ is added to keep the term non-zero.

## 2.2 Boundary conditions

The upper surface is assumed to be traction free. The ice sheet base is subject to basal sliding according to a friction law for the basal shear stress $t_{\mathrm{b}}^{\mathrm{D}}$ in the tangential plane

$$t_{\mathrm{b}}^{\mathrm{D}} = \tau_{\mathrm{b}} = [(\boldsymbol{I} - \boldsymbol{n}\boldsymbol{n}^T)t^{\mathrm{D}}] \cdot \boldsymbol{n} = -\beta^2(\boldsymbol{I} - \boldsymbol{n}\boldsymbol{n}^T) \cdot \boldsymbol{v} = -\beta^2\boldsymbol{v}_{\mathrm{b}}, \tag{6}$$

$$\boldsymbol{v} \cdot \boldsymbol{n} = 0, \tag{7}$$

with the unit normal vector $\boldsymbol{n}$ pointing out of the ice, and $\boldsymbol{v}_{\mathrm{b}}$ is the velocity in the tangential plane at the base. Basal refreezing or melting is neglected. The basal drag parameter $\beta^2$ often includes the effective pressure to account for the presence of water lubricating the ice-bed interface, as well as bed properties such as roughness. For floating ice we use the same boundary condition but prescribe free slip ($\beta^2 = 0$). The parameterization for $\beta^2$ and the remaining lateral boundary condition are given below for each experiment setup.

## 2.3 Model classification

The mass and momentum balance (Eq. 1 and 2) can be written in terms of velocity components and pressure $p$

$$\frac{\partial v_x}{\partial x} + \frac{\partial v_y}{\partial y} + \frac{\partial v_z}{\partial z} = 0,$$

$$\frac{\partial}{\partial x}\left(2\eta\frac{\partial v_x}{\partial x}\right) + \frac{\partial}{\partial y}\left(\eta\frac{\partial v_x}{\partial y} + \eta\frac{\partial v_y}{\partial x}\right) + \frac{\partial}{\partial z}\left(\eta\frac{\partial v_x}{\partial z} + \eta\frac{\partial v_z}{\partial x}\right) - \frac{\partial p}{\partial x} = 0,$$

$$\frac{\partial}{\partial x}\left(\eta\frac{\partial v_x}{\partial y} + \eta\frac{\partial v_y}{\partial x}\right) + \frac{\partial}{\partial y}\left(2\eta\frac{\partial v_y}{\partial y}\right) + \frac{\partial}{\partial z}\left(\eta\frac{\partial v_y}{\partial z} + \eta\frac{\partial v_z}{\partial y}\right) - \frac{\partial p}{\partial y} = 0,$$

$$\frac{\partial}{\partial x}\left(\eta\frac{\partial v_x}{\partial z} + \eta\frac{\partial v_z}{\partial x}\right) + \frac{\partial}{\partial y}\left(\eta\frac{\partial v_y}{\partial z} + \eta\frac{\partial v_z}{\partial y}\right) + \frac{\partial}{\partial z}\left(2\eta\frac{\partial v_z}{\partial z}\right) - \frac{\partial p}{\partial z} = -\varrho_i g, \tag{8}$$

as well as the effective strain rate occurring in Eq. 5

$$\dot{\varepsilon}_e = \left(\frac{1}{2}\left(\frac{\partial v_x}{\partial x}\right)^2 + \frac{1}{2}\left(\frac{\partial v_y}{\partial y}\right)^2 + \frac{1}{2}\left(\frac{\partial v_z}{\partial z}\right)^2 + \frac{1}{4}\left(\frac{\partial v_x}{\partial y} + \frac{\partial v_y}{\partial x}\right)^2 + \frac{1}{4}\left(\frac{\partial v_x}{\partial z} + \frac{\partial v_z}{\partial x}\right)^2 + \frac{1}{4}\left(\frac{\partial v_y}{\partial z} + \frac{\partial v_z}{\partial y}\right)^2\right)^{1/2}. \tag{9}$$

The red terms are omitted in the BP approximation as explained below. The four equations (Eqs. 1 and 8) with the four unknowns $v_x$, $v_y$, $v_z$ and $p$ form the FS equation system. Various approximations to this set of equation exist ranging from shallow ice approximation (SIA) to different types of one layer and multilayer approximations for non-horizontal shear components of the stress tensor (Hutter, 1983; Hindmarsh, 2004; Bueler and Brown, 2009; Ahlkrona et al., 2013). The FS problem has saddle point character which is difficult to solve (Benzi et al., 2005). In order to reduce the computational demand and derive a better posed problem, Blatter (1995) and Pattyn (2003) developed the so-called BP approximation scheme which is valid

for most of the ice sheet domain and widely used in ice flow models. The main assumption of their model is that the vertical component of the momentum balance is approximated as hydrostatic (i.e. $\partial t_{zz}/\partial z = \varrho_i g$). This assumption is used to eliminate the pressure variable $p$ from the FS equation system and reduces the FS problem with the two horizontal velocity components as unknown field variables. The further assumption that horizontal gradients of the vertical velocity are small compared to the vertical gradient of the horizontal velocity (i.e. $\partial v_z/\partial x \ll \partial v_x/\partial z$ and $\partial v_z/\partial y \ll \partial v_y/\partial z$) leads to a closed problem for the two horizontal velocity components. This approximation is equivalent to dropping the red coloured terms in Eq. 8 and 9; it is termed LMLa in Hindmarsh (2004).

The assumptions to the BP scheme imply that the so-called bridging stresses (also known as vertical resistive stress, van der Veen and Whillans, 1989), i.e. the resistance to varying stress gradients in direction of the ice flow, is neglected. Bridging effects are generally small and occur near the ice divide and at the grounding line (Pattyn, 2000). Since the BP schemes retain stresses of $\mathcal{O}(1)$ and $\mathcal{O}(\epsilon)$ but delete stress terms of $\mathcal{O}(\epsilon^2)$ (Blatter, 1995) it is only valid to a certain aspect ratio and topographic variability. Once high velocity gradients develop over short distances BP may not provide an accurate solution.

The simplifications made by BP on the FS system reduces the computational demand by deriving a better posed problem. By re-arranging the BP equations this system leads to a closed problem for the two horizontal velocity components $v_x$ and $v_y$. In doing so, the BP approximations decouple the vertical velocity $v_z$ and the pressure $p$ from the full system. Both unknowns are recovered diagnostically by integrating the incompressibility equation (Eq. 1) and the vertical component of the momentum balance equation (Eq. 8), respectively. However, in our approach we follow an alternative way to align with the BP stress regime, thereby directly allowing to compare FS and BP model simulations (see Sect. 3.1 and 3.2).

## 3   Model description and verification

### 3.1   Ice flow model

To solve the set of equations presented above (Eq. 1 and 2 and boundary conditions), we use the COMmercial finite element SOLver (COMSOL) Multiphysics© (version 5.5, www.comsol.com, last access May 2021). COMSOL provides a FS interface which is additionally manipulated in order to align with the BP stress regime. This is accomplished by discarding the corresponding terms (red terms in Eqs. 8 and 9) in the weak formulation but maintaining identical FS numerical details. Through this approach the BP stress regime does not further modify the default FS module, i.e the BP version implemented in this study is not forming a multilayer model with the two horizontal velocity components as field variables. Therefore, our implemented BP approximation is hereafter termed BP-like. In doing so, both flow modes (FS and BP-like) are running through the same (FS) solving algorithm. Hereby we ensure that differences in the computed results due to numerical issues are largely eliminated. However, the disadvantage is that we do not benefit by the reduced complexity of the BP approximation. As a consequence of the avoided decoupling $\partial v_z/\partial z$ in Eq. 9 is not replaced by $-(\partial v_x/\partial x + \partial v_y/\partial y)$ as commonly done in HO approximations (e.g., Pattyn, 2003).

## 3.2 Numerics

The mass balance and momentum equations constitute a nonlinear equation system. Independent of the employed flow regime,
a damped Newton method (which is COMSOL's default solver) is applied in a fully coupled manner to solve for the velocity
vector $v$ and the pressure $p$. The unstabilized Stokes equation (Eq. 8) is subject to the Babuska-Brezzi condition, which states
that the basis functions for $p$ have to be of lower order than for $v$. This issue could be circumvented with suitable stabilization
techniques. Here, we test the numerical robustness of our results by employing two different discretization schemes: (1) We
use P2+P1 Lagrange elements which consists of quadratic basis functions for the velocity and linear for the pressure. (2) The
second scheme uses linear elements for both the velocity components and the pressure field (P1+P1 Lagrange elements) and
numerical stabilization is achieved with streamline diffusion (Galerkin least-squares (GLS), Hauke and Hughes, 1994). Due to
the equal-order interpolation, the latter is computationally more efficient but less accurate and sensitive to the GLS stabilization
parameters (Helanow and Ahlkrona, 2018). In the following, the schemes will be referred as P2P1 and P1P1GLS, respectively.

To further test the numerical robustness we compare two different implementations (called 'strong' and 'weak') for the
Dirichlet condition (Eq. 7) at the ice base. The strong imposition applies the constraints pointwise at each node and, therefore,
modifies the underlying variational formulation. The weak method is rather a component of the variational formulation by
adding exterior facet integrals. For the latter, we use the so-called Lagrange multiplier method (Babuska, 1973; Verfürth, 1986;
Urquiza et al., 2014) where the Dirichlet condition is multiplied with the Lagrange multiplier $\lambda$.

The flow regime equations are forming a saddle point problem, for which either a direct solver or special preconditioning
is needed. The verification tests ISMIP-HOM (see Sect. 3.3) are solved with the direct solver MUMPS (https://graal.ens-lyon.
fr/MUMPS/; last access January 2022). For the NEGIS region experiments (Sect. 4), the degrees of freedom (DOF) increase
rapidly with increasing spatial resolution, and a direct solver would be uncommon for such large numbers of DOFs. For the
larger problems we rely on the iterative GMRES solver (Saad, 2003) which is accelerated with appropriate preconditioners.
Simulations that make use of the strong imposition rely on a Domain Decomposition solver with an overlapping additive
Schwarz method (ASM, Widlund and Toselli, 2005) which requires less computation time and working memory compared to
MUMPS. Unfortunately, the ASM preconditioner shows a very high computational demand for simulations that employ the
weak imposition. Since the involved Lagrange multiplier induces a zero on the diagonal of the system matrix, several common
preconditioners are not applicable. Therefore, we employ the Vanka algorithm (Vanka, 1986; John and Matthies, 2001) which
is specifically designed for large indefinite problems with saddle point character. Based on our simulations, we found Vanka to
be very memory efficient and computationally fast for large problems although it requires more Newton and GMRES iterations
compared to ASM (see Sect. 4 and Tab. 1 and 2).

Regarding the desired consistency between the simulation results, we must clarify the different usage of the linear solvers:
the coarser models show a poor performance with the ASM and Vanka preconditioner. At the same time, the higher resolutions
run out of memory with the MUMPS solver. Depending on modelling domain and employed discretization, MUMPS runs
out of memory around 5 million DOF's. For the largest DOF model that could be solved with the direct solver MUMPS on
our cluster system, we also performed a simulation with the iterative linear solvers ASM and Vanka. A comparison of both

simulations reveals differences well below $10^{-7}\,\mathrm{m\,a^{-1}}$ in the surface velocity, $v_s = |\boldsymbol{v}|_s$; differences between MUMPS and Vanka are below $10^{-3}\,\mathrm{m\,a^{-1}}$. Consequently, we assume that all applied linear solvers provide comparable results.

### 3.3 Verification by benchmarks

The ISMIP-HOM benchmark targets comparison for three dimensional ice flow models arising from basal topography undulations, from basal sliding and by changing the length scale. From the ISMIP-HOM setup (Pattyn et al., 2008) we select the 3D and non-transient experiments Exp. A ('Ice flow over a bumpy bed') and Exp. C ('Ice stream flow I'). We rely on this setup to verify our flow implementations, in particular the BP-like implementation. Exp. A is a parallel-sided slab with a series of sinusoidal oscillations in the basal topography with a frequency of $\omega = 2\pi/L$, with $L$ being the domain length. In this setup no basal sliding is prescribed (i.e. $\boldsymbol{v}|_b = 0$). Exp. C is a parallel-sided slab with a series of sinusoidal oscillations on the basal drag parameter $\beta^2$ (Eq. 6) with the frequency $\omega$. At the lateral margins, periodic boundary conditions are applied. The length scale is subsequently reduced as $L =$160, 80, 40, 20, 10 and 5 km so that full-Stokes effects can develop due to the higher aspect ratio $H/L$, with $H$ being the mean ice thickness. The domain is discretized with $n_x = n_y = L/41$ grid points in the horizontal plane. In the vertical direction, 15 layers refined to the base are chosen. We run the benchmarks with a length scale of 5 km with the linear (P1P1GLS) and second order (P2P1) element functions. Additionally, Exp. C is run with the strong and weak imposition of the basal friction law. All other length scales are run with P1P1GLS and for Exp. C with the strong imposition of the friction law. The ISMIP-HOM experiments use an ice density of $\varrho_i = 910\,\mathrm{kg\,m^{-3}}$. For a detailed description of the experimental setup, please refer to Pattyn et al. (2008).

Our results for FS and BP-like show good agreements with the corresponding models for FS and non FS (nFS; nFS includes BP) for the conducted A and C test cases (see Figs. S2 and S3). However, the original exercise reveals distinct differences between FS and all other approximations on short length scales (Fig. 1). In particular, for Exp. C the 'double-peak' in the surface velocity is present in the FS solution but absent in the BP-like regime, which is consistent with the original ISMIP-HOM results. Independent of the used element functions and friction law imposition, we found a good agreement with the original ISMIP-HOM contributions and within our own model versions. Generally, the P2P1 model reveals slightly higher surface velocities than P1P1GLS. The different friction law implementations show very minor differences. From the resulting behavior, the implementations of the FS and BP-like stress regimes are verified.

## 4 Description of NEGIS region experiments

We select two different regions of the NEGIS (Fig. 2) to investigate the effect of FS and BP on its ice dynamics. The first region focuses on an ice-stream like flow regime with abrupt changes in slow and fast flow of the southern branch (SB) of NEGIS (it is termed 'ice-stream', see red line in Fig. 2). The second region also includes the western branch (WB) of NEGIS and the outlet regions of 79NG and ZI (it is termed 'outlet', see cyan line in Fig. 2). For the ice-stream experiment a region upstream of the grounding line and downstream of the ice divide is selected to avoid grounding line treatment and to focus on an ice-stream like flow regime. The outlet region captures the grounding zone of 79NG and the marine terminus of ZI. As we

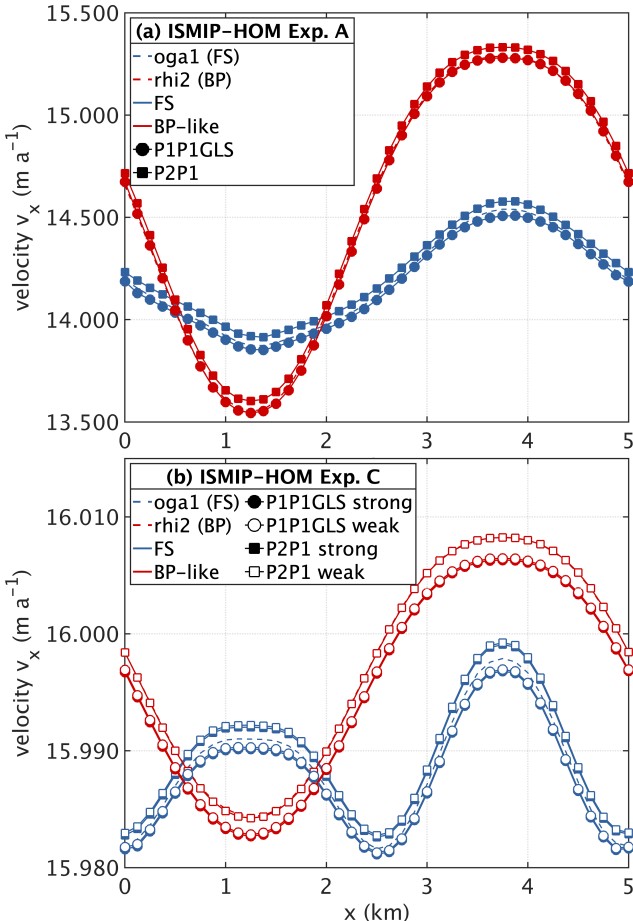

**Figure 1.** Results of the ISMIP-HOM experiments A (a) and C (b) for the length scale $L = 5\,\text{km}$. Surface velocity component $v_x$ $(\text{m a}^{-1})$ at $y = L/4$. Values computed in this study for FS and BP-like are compared to the BP model 'rhi2' and the FS model 'oga1' from the original ISMIP-HOM benchmark (Pattyn et al., 2008). Please note that the BP-like P1P1GLS and 'rhi2' BP solutions in Exp. A overlap on each other; in Exp. C the 'rhi2' BP solution is overlaid by BP-like P2P1 strong and and BP-like P2P1 weak.

rely on the BedMachine v3 data-set (BM, Morlighem et al., 2017), a small area at 79NG becomes afloat. In both setups, the
geometry of the NEGIS regions is laterally defined along flow lines that were deduced from the MEaSUREs surface velocity
data-set (Joughin et al., 2016, 2018).

Bed and surface topography are taken from BM and the GIMP data set (Howat et al., 2014), respectively. As consistency is
the primary target of the model set-up to study the response of both stress regimes to different initial and boundary conditions,
we do not perform an initial relaxation of the free surfaces. A relaxation run based on e.g., the BP-like scheme and subsequent
simulation runs with relaxed geometry with BP-like and FS would likely experience a 'shock' (Goelzer et al., 2018) for
the FS model because the geometry is consistent to the BP-like physics (and in the same way also for a FS relaxation run).
Performing relaxation simulations for BP-like and FS individually would result in different initial geometries and also different

initial conditions (e.g., velocity). This would be an undesired effect for our consistency approach which would make a clear comparison difficult. Here, we followed the strategy to have physical parameters and the geometry input equal in the FS and

BP-like to allow for a better comparison.

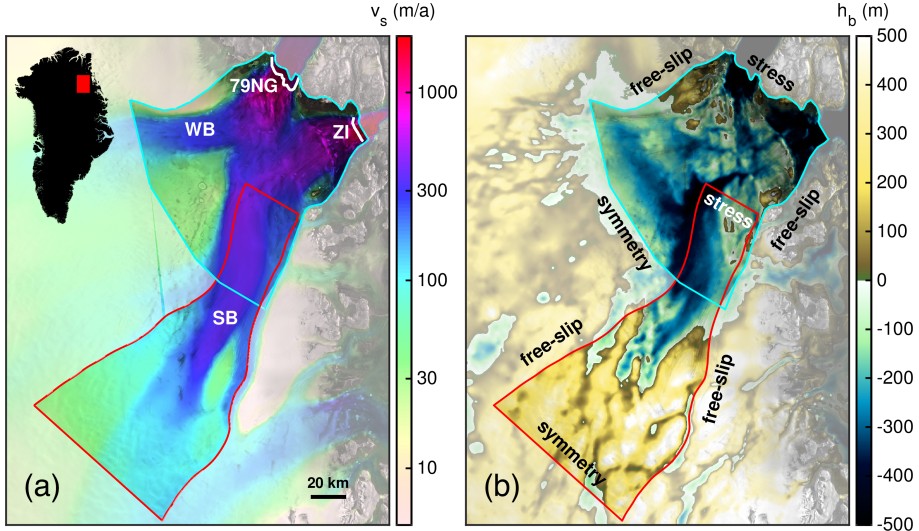

**Figure 2.** Overview of the NEGIS region and modelling domain. (a) Observed surface velocities (Joughin et al., 2016, 2018). (b) Bed topography (Morlighem et al., 2017). The red and cyan line in (a) and (b) delineates the ice-stream and outlet modelling domain, respectively. White lines in (a) indicate flux gate locations at 79NG and ZI (see Fig. 9 and related text below). WB and SB indicate the western and southern branch of the NEGIS. Symmetry, free-slip, and stress in (b) indicate the boundary condition types used a lateral margins. Background image is a RADARSAT Mosaic (Joughin, 2015; Joughin et al., 2016)

In the NEGIS experiments we consider a Budd-like friction law (Budd et al., 1979) with a linear ($m = 1$) and non-linear relationship ($m = 3$). The basal drag parameter $\beta^2$ in the basal boundary condition (Eq. 6) is assumed to be:

$$\beta^2 = k^2 N^{1/m} \boldsymbol{v}_{\mathrm{b}}^{1/m-1}, \tag{10}$$

where $k$ is the basal friction coefficient found by an inversion technique (see below and Appendix A). The effective normal

pressure is the difference between the ice pressure $P_{\mathrm{i}}$ and the basal water pressure $P_{\mathrm{w}}$

$$N = P_{\mathrm{i}} - P_{\mathrm{w}}. \tag{11}$$

The basal water pressure is computed in marine parts, i.e. where the ice base is below the sea-level ($h_{\mathrm{b}} < z_{\mathrm{sl}}$), according to Huybrechts (1992)

$$P_{\mathrm{i}} = -(\boldsymbol{t} \cdot \boldsymbol{n}) \cdot \boldsymbol{n}, \tag{12}$$

$$P_{\mathrm{w}} = -\min(\varrho_{\mathrm{w}} g h_{\mathrm{b}}, 0), \tag{13}$$

where $\varrho_{\mathrm{w}} = 1023\,\mathrm{kg\,m^{-3}}$ is the density of the ocean water and $h_{\mathrm{b}}$ the bed elevation. Here, $P_{\mathrm{i}}$ is the normal stress instead of the approximated ice overburden pressure. The assumptions made about the water pressure imply that the base is perfectly connected to the ocean at any location in the domain that is below sea-level. Overall, this might be incorrect but it forms an appropriate assumption in the absence of an additional hydrological model.

Solving a subset of an ice stream poses unknown boundary conditions in the interior of the ice sheet. A simple approach would be prescribing the measured surface velocities as a depth-averaged velocity profile. However, we choose boundary conditions that are free to adjust during the solution process (Fig. 2b). Inflow boundaries are chosen to have a symmetry boundary condition, which implies $\boldsymbol{v} \cdot \boldsymbol{n} = 0$ and vanishing shear stresses. A free-slip condition is chosen at lateral along-flow boundaries and a normal stress condition at the outflow boundaries. Dependent on the setup, the outflow boundaries are land-

or marine terminating fronts of the glacier or located within the ice. For those outflow boundaries we chose stress boundary condition $\boldsymbol{t} \cdot \boldsymbol{n} = -F_0 \boldsymbol{n}$. For outflow boundaries located within the ice (e.g., at 79NG) we choose $F_0 = \varrho_i g(z_s - z) + \delta$; for land- or marine terminating fronts (e.g., ZI) we set $F_0 = \min(\varrho_i g(z_s - z), 0) + \delta$, where $z_s$ is the surface elevation and $z$ the vertical coordinate. Here, $\delta$ is an additional stress that is tuned to match the observed velocities at those boundaries. The additional stress is set to zero except at 79NG and ZI; where $\delta$ is 0.5 MPa and 0.3 MPa at 79NG and ZI, respectively.

Dependent on the setup, we conduct experiments ranging from a horizontal mesh resolutions of $l = 12.8$ to 0.1 km. Please note, that $l = 0.15$ km is the nominal resolution of the BM data-set. The horizontal grid sizes span a typical range of resolutions used in ice sheet projections (e.g., Aschwanden et al., 2019; Goelzer et al., 2020; Seroussi et al., 2020) and beyond. The 3D meshes comprise between 1960 and 32 864 970 prism elements, respectively, resulting in 5280 to 72 416 740 DOF for the mesh cases. For a number of simulations, the enhancement factor in Eq. 5 is varied as $E = 0.1, 1, 3$, and 6 to investigate whether

simulation differences stem from basal sliding (small $E$) or internal velocity (large $E$). In the following $E = 1$ will be referred as the 'classical case'. The upper and lower values of $E$ are extreme but not too far away from currently used values (Ma et al., 2010; de Boer et al., 2015). To test the sensitivity on the assumed friction law we run simulations with $m = 1$ and $m = 3$ (Eq. 10). The numerical robustness is tested by the aforementioned numerical discretization schemes and the implementations of the basal boundary condition. We do not run all possible combination in all setups as it is computationally too expensive.

Overall, computational times varied between a few seconds and several hours. A broad overview of computational costs is given in Tab. 1 and 2. If not mentioned otherwise all setups employ 11 vertical layers (refined to the base) which is sufficient compared to higher resolutions (Fig. S1).

    For the parameters $A(T, W)$ (Eq. 5) and $k^2$ (Eq. 10) we utilize external products and capacities:

1. To rely on a realistic thermal state but avoiding an intensive thermal spin-up, we make use of an enthalpy field from

a paleo-climatic spin-up simulated by the ice sheet model SICOPOLIS with SIA (Rückamp et al., 2019). The field is bilinearly interpolated to the COMSOL meshes. The temperature-dependent part of the rate factor in Eq. 5 for cold ice is computed following Cuffey and Paterson (2010, Tab. 3.4 therein), and the water-content-dependent part for temperate ice is computed following Lliboutry and Duval (1985).

**Table 1.** Overview of mesh characteristics and consumed computational resources of the ice-stream region. Number of non-linear iterations (damped Newton's method), computational time, and physical RAM usage are listed for $E = 0.1$, $m = 1$, FS and P1P1GLS strong; values do not vary much among other $E$ and stress regimes settings. Solver and cluster settings (OpenMP (OMP) shared-memory threads and distributed-memory tasks (MPI)) are kept constant between each resolution and stress regime. Computational time includes the solver procedure (e.g. assembling the stiffness matrix, load vector) and outside routines (e.g. construction of mesh, saving results). Generally, COMSOL is run in hybrid mode, meaning the process is distributed to a user-defined number of OMP threads and MPI tasks. Without testing the performance thoroughly we achieve a good performance by OMP parallelism per physical node and MPI communication between physical nodes (the code is simply run in OMP mode if MPI is equal to one). The listed RAM usage refers to one MPI task. Please note, that the cluster settings (MPI and OMP) have been chosen in a way that the job returns a solution within the cluster 'wall-time' but do not indicate any performance scaling.

| resolution (km) | #elements | #DOF | lin-Solver | #nlin \| lin | wall-time (min) | RAM (GB) | MPI \| OMP |
|---|---|---|---|---|---|---|---|
| 12.8[†] | 1960 | 5280 | MUMPS | 11 \| - | 0.11 | 1.87 | 1 \| 36 |
| 6.4[†] | 8560 | 20680 | MUMPS | 10 \| - | 0.48 | 2.24 | 1 \| 36 |
| 3.2[†] | 32700 | 75548 | MUMPS | 9 \| - | 1.72 | 4.03 | 1 \| 36 |
| 1.6[†] | 128530 | 289960 | MUMPS | 9 \| - | 2.99 | 4.80 | 4 \| 36 |
| 0.8[†] | 512640 | 1142108 | MUMPS | 9 \| - | 15.22 | 12.29 | 4 \| 36 |
| 0.4[†] | 2066130 | 4573976 | ASM* | 10 \| 45 | 86.50 | 28.91 | 24 \| 18 |
| 0.2[†] | 8220400 | 18141816 | ASM* | 10 \| 46 | 248.75 | 46.95 | 24 \| 9 |
| 0.15[†] | 14556150 | 32099408 | ASM* | 11 \| 50 | 320.75 | 47.15 | 56 \| 6 |
| 0.1[‡] | 32864970 | 72416740 | ASM* | 11 \| 57 | 708.11 | 359.60 | 30 \| 48 |

[†] Simulations are run on the AWI cluster Cray CS 400. Each production node is equipped with 64 GB RAM and 2x18-Core CPUs (Intel Xeon Broadwell CPU E5-2697).

[‡] Simulations are run at the North-German Supercomputing Alliance (HLRN) cluster Lise. Each production node has 362 GB and 2x48-Core CPUs (Intel Cascade Lake Platinum 9242 (CLX-AP).

* Domain Decomposition solver with an overlapping multiplicative Schwarz method. Number of sub-domains are set equally to the number of MPI tasks.

2. We make use of the inversion capability in the Ice-sheet and Sea-level System Model (ISSM, Larour et al., 2012) to obtain a spatially varying basal friction coefficient $k^2$ (Fig. A1 and A2, see Appendix A). The ISSM inversion is performed with $l = 0.8$ km, with BP and for the individual $E$ values. Similarly, the SICOPOLIS temperature field is interpolated to the 3D ISSM mesh. Subsequently, the inferred fields for $k^2$ are bi-linear interpolated to the COMSOL meshes.


**Table 2.** Overview of mesh characteristics and consumed computational resources of the outlet region similar as in Tab. 1. Number of non-linear iterations (damped Newton's method), computational time, and physical RAM usage are listed for $E = 1$, $m = 3$, FS and P1P1GLS weak.

| resolution (km) | #elements | #DOF | lin-Solver | #nlin \| lin | wall-time (min) | RAM (GB) | MPI \| OMP |
|---|---|---|---|---|---|---|---|
| 12.8 | 2620 | 6820 | MUMPS | 15 \| - | 0.1 | 2.19 | 1 \| 36 |
| 6.4 | 8440 | 20284 | MUMPS | 14 \| - | 0.3 | 2.86 | 1 \| 36 |
| 3.2 | 33870 | 77792 | MUMPS | 14 \| - | 0.75 | 4.78 | 1 \| 36 |
| 1.6 | 133380 | 299860 | MUMPS | 12 \| - | 1.86 | 5.27 | 4 \| 36 |
| 0.8 | 536220 | 1192400 | MUMPS | 12 \| - | 6.93 | 12.58 | 4 \| 36 |
| 0.4 | 2125600 | 4701708 | Vanka | 12 \| 842 | 21.75 | 13.98 | 8 \| 36 |
| 0.2 | 8494560 | 18738676 | Vanka | 13 \| 2854 | 88.48 | 33.24 | 16 \| 36 |
| 0.15 | 15024320 | 33121000 | Vanka | 13 \| 2461 | 102.03 | 47.37 | 32 \| 18 |

All simulations are run on the AWI cluster Cray CS 400. Each production node is equipped with 64 GB RAM and 2x18-Core CPUs (Intel Xeon Broadwell CPU E5-2697).

## 5 Results of the NEGIS region experiments

### 5.1 Ice-stream region

An overview of input parameters and FS simulation results of the ice-stream region for $l = 6400\,\mathrm{m}$ and $100\,\mathrm{m}$ is provided in Fig. 3. The slope of the bed topography (Fig. 3a, d) and the friction coefficient (Fig. 3b, e) of the finer resolution reveals more small scale features and stronger amplitudes; particularly a rapidly varying bed. Despite both compared resolutions relying on the same inferred basal friction coefficient and input geometry, the interpolation to the computational grids produces striking differences. As a consequence the simulated surface velocity fields differ quantitatively (Fig. 3c, f). Both resolutions reproduce

the NEGIS to some degree but the finer resolution features an extended fast flow region ($\gtrsim 350\,\mathrm{m\,a^{-1}}$) and more pronounced shear margins (i.e., sharper velocity gradients). As the ISSM simulations inferring the friction coefficient are conducted with a 0.8 km horizontal grid resolution and with the BP ice flow approximation, the COMSOL results among all applied resolutions and flow regimes are not expected to match the observed velocity field. The purpose of the transferred friction coefficient from the ISSM inversion to COMSOL is to roughly reproduce the observed NEGIS flow pattern. A better fit to observations could

certainly be achieved by performing the inversion for each setting separately; however, our experiments are designed to focus on stress regime inter-comparisons and not on an assessment to observations.

In the following, the FS solution is considered as the exact solution. Therefore, our analysis builds on presenting differences of the BP-like solution to FS. The differences of surface velocities, $\Delta v_s^{\mathrm{FS-BP\text{-}like}} = v_s^{\mathrm{FS}} - v_s^{\mathrm{BP\text{-}like}}$, between FS and BP-like are displayed for the $100\,\mathrm{m}$ grid resolution in Fig. 4. Spatial differences reveal a different pattern among the $E$ values. For stiffer

ice ($E = 0.1$) the BP-like solutions exhibit higher flow speeds which is almost confined to the fast flow region (Fig. 4a); this

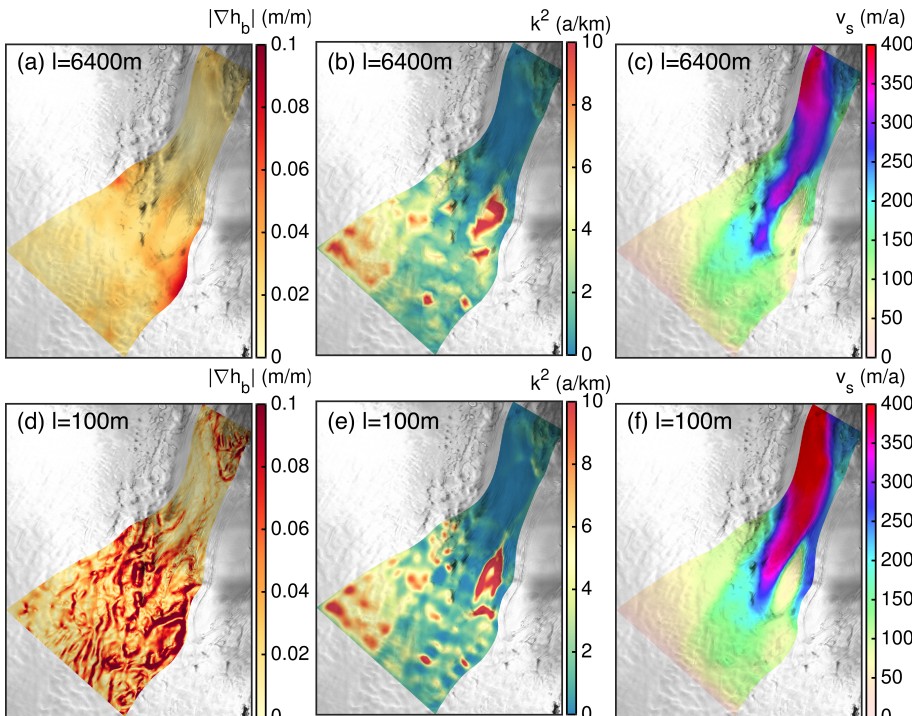

**Figure 3.** Overview of input parameters and exemplary simulation results of the ice-stream NEGIS region. Upper row shows slope of bed topography (a), basal friction coefficient for $E = 1$ inferred with ISSM (b), and simulated surface velocity with FS, $E = 1$, $m = 1$ and with FS P1P1GLS-strong (c) for 6400 m horizontal resolution. Lower rows shows the same fields but for 100 m horizontal resolution. Background image is a RADARSAT Mosaic (Joughin, 2015; Joughin et al., 2016).

area partly coincides with a smooth bed and surface topography (see Fig. 3). Overall, discrepancies between FS and BP-like increase with higher velocities. The largest impact of using a BP-like solution results from the surface flow velocities, which are up to 43 m a$^{-1}$ faster (relative difference is about 19%) compared to FS. For the 'classical case' ($E = 1$) the pattern is similar but with reduced magnitudes. In this setup, the BP-like velocities are up to 19 m a$^{-1}$ higher (relative difference is about 290 6%). With softer ice ($E > 1$), the pattern in surface velocity differences changes and, eventually, for very soft ice ($E = 6$) differences are more pronounced in regions with higher topographic variability (compare Fig. 3) and exhibit a rippled pattern. Velocity differences are scattered around the FS solution by $\pm 80$ m a$^{-1}$; the spread in differences is more pronounced for softer ice (see scatter plots in Fig. 4).

Based on the spatial differences, we compute a spatially-averaged surface velocity for all parameter settings to obtain an 295 integrative overview (Fig. 5). The analysis generally reveals that both the absolute spatially-averaged surface velocity (Fig. 5a– d) and relative spatially-averaged surface velocity differences (Fig. 5e–h) start to diverge somewhere below 1.6 and 0.8 km horizontal grid resolution (which is in the range of the mean ice thickness of 1330 m). Away from this threshold towards finer resolutions, solution differences increase between the stress regimes. For the finer resolutions BP-like produces higher mean

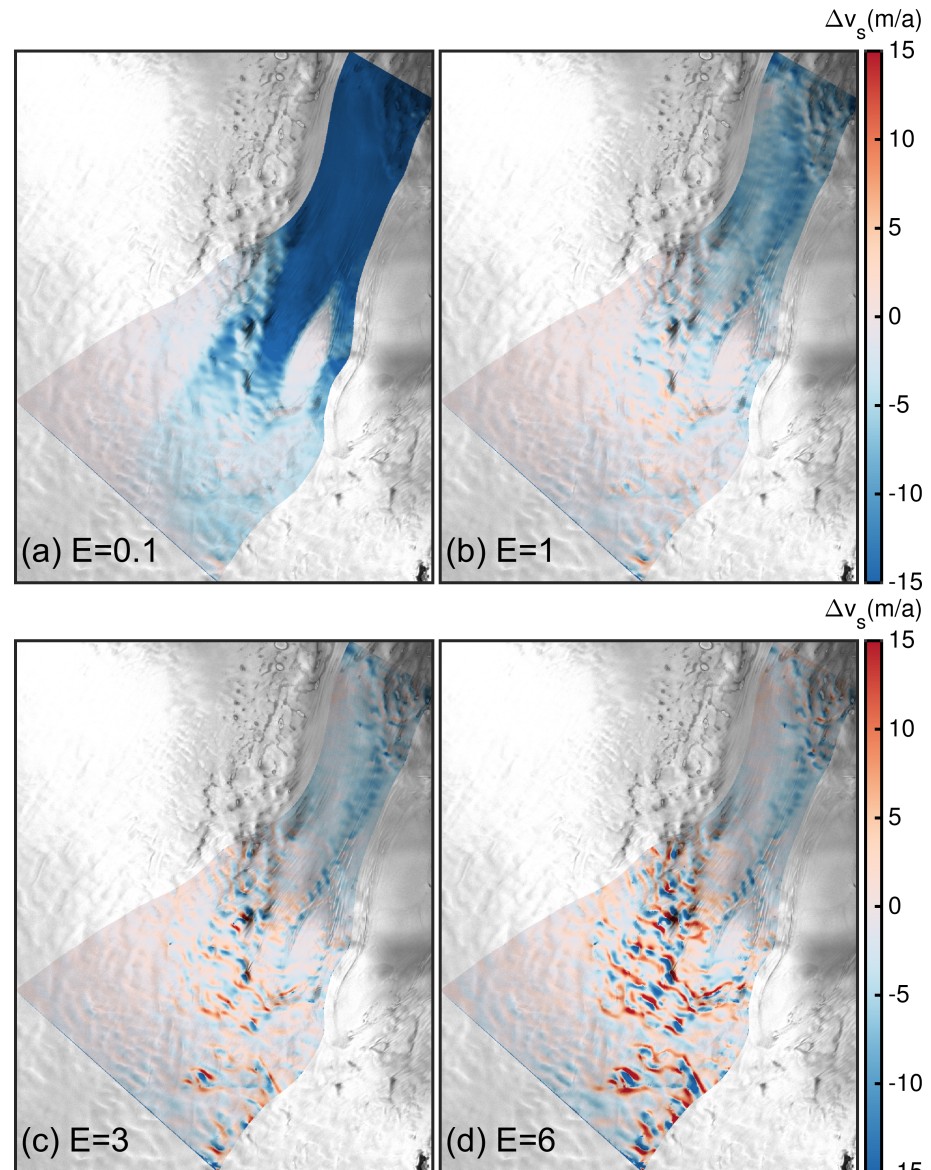

**Figure 4.** Difference of surface velocity $\Delta v_s^{\text{FS}-\text{BP-like}} = v_s^{\text{FS}} - v_s^{\text{BP-like}}$ for a resolution of 100 m. (a, b, c, d) Differences for $E = 0.1$, 1, 3 and 6, respectively. Background image is a RADARSAT Mosaic (Joughin, 2015; Joughin et al., 2016).

surface velocities compared to FS. At the finest resolution (below the nominal resolution of the BM data-set), the mean velocity seem to reach a converged state for the BP-like stress regime (less pronounced for $E = 0.1$), but FS reveals a small dip towards lower velocities. The tendency of the diverging relative differences at the finest resolution is therefore likely a result from the still changing FS solution under grid refinements. The obtained tendency below 150 m is intriguing as the differences solely arise from grid-refinements and not due to the input geometry.

Relative maximum differences between FS and BP-like are obtained for the stiff-ice case ($E = 0.1$). A discussion on why
stiff ice is more sensitive will come in section 5.5. The spatially-averaged ice flow velocity difference is up to $7\,\mathrm{m\,a^{-1}}$ for
$E = 0.1$; the relative error between FS and BP-like is up to $\sim 5.8\%$. The resulting difference for $E = 1$ shows the same trend
but is of lower magnitude (up to $\sim 1.5\%$). With softer ice ($E > 1$) the differences decrease. Interestingly, both stress regimes
reveal an intermediate peak in mean surface velocity at $l = 1600\,\mathrm{m}$ over all $E$ values.

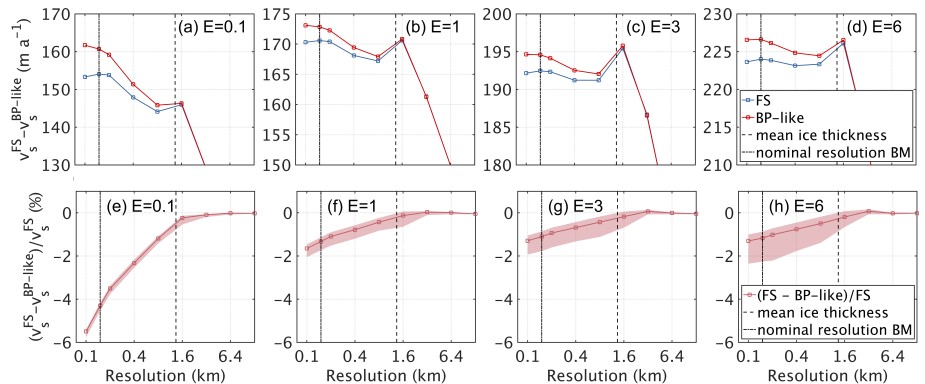

**Figure 5.** Integrative overview of spatially averaged surface velocities from FS and BP-like stress regimes. Upper row (a, b, c, d) shows surface velocity mean for $E = 0.1$, 1, 3, and 6, respectively. Lower row (e, f, g, h) shows the relative difference of BP-like to FS for $E = 0.1$, 1, 3, and 6, respectively. The lighter background indicates the local maximum and minimum relative difference. Please note the logarithmic scale of the x-axis and the different scale of the y-axis in the upper row.

## 5.2 Outlet region

As for the ice stream region, an overview of input parameters and FS simulation results of the outlet NEGIS region are
compared for $l = 6400\,\mathrm{m}$ and $150\,\mathrm{m}$ with P1P1GLS-weak and $m = 3$ (Fig. 6). Similarly, the slope of the bed topography
(Fig. 6a, d) and the friction coefficient (Fig. 6b, e) of the finer resolution reveals more small scale features and stronger
amplitudes. In both resolutions, the simulated surface velocity fields agree fairly well and reproduce the NEGIS (Fig. 6c, f).
However in shear margins and gradients in along flow direction are more pronounced in higher resolution.
Differences of surface velocities between FS and BP-like are displayed for the $150\,\mathrm{m}$ grid resolution, $E = 1$ and P1P1GLS in
Fig. 7. Overall, spatial differences reveal a similar pattern among the employed friction exponents and friction implementations.
Largest differences occur in the vicinity of the 79NG grounding line (up to $-570\,\mathrm{m\,a^{-1}}$, see Tab. 3). For a power friction law
($m = 3$) the differences are larger compared to the linear friction law ($m = 1$). The P1P1GLS-weak setup produces slightly
larger differences than P1P1-strong.
We compute relative spatially-averaged surface velocity differences for all parameter settings to obtain an integrative overview
(Fig. 8). The analysis discloses the clear trend that BP-like simulations lead to higher velocities than FS on average. Overall,
the differences increase with higher resolution independent of the employed numerical characteristics. Interestingly, differ-

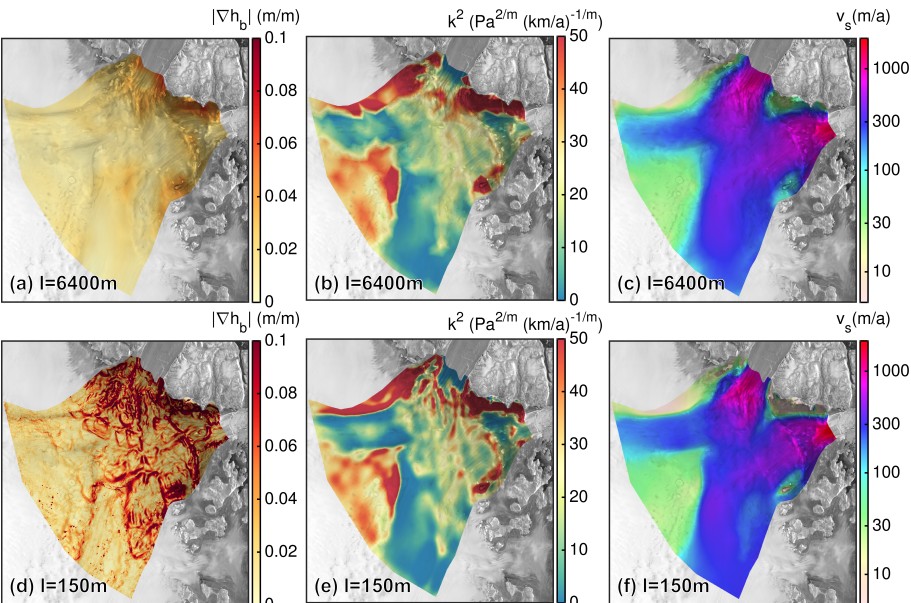

**Figure 6.** Overview of input parameters and simulation results of the outlet NEGIS region. Upper row shows (a) slope of bed topography, (b) basal friction coefficient for $E = 1$, and (c) simulated surface velocity with FS P1P1GLS-weak and $m = 3$ for 6400 m horizontal resolution. Lower rows shows the same fields but for 150 m horizontal resolution. Background image is a RADARSAT Mosaic (Joughin, 2015; Joughin et al., 2016).

**Table 3.** Maximum absolute velocity difference ($\max(|\Delta v_s^{\text{FS}-\text{BP-like}}|)$, unit is $\text{m a}^{-1}$) and their corresponding relative velocity difference ($\max(|\Delta v_s^{\text{FS}-\text{BP-like}}|)/v_s^{\text{FS}}$, in brackets, unit is %) for different areas in the outlet domain. Results are listed for $l = 150$ m, P1P1GLS and different setups for the basal boundary condition.

| setup | 79NG | ZI | SB | WB |
|---|---|---|---|---|
| strong, $m = 1$ | 353.8 (36.1) | 133.9 (10.8) | 42.7 (14.2) | 27.0 (10.0) |
| strong, $m = 3$ | 435.9 (47.4) | 154.6 (12.4) | 70.1 (24.6) | 45.5 (15.6) |
| weak, $m = 1$ | 491.3 (53.7) | 161.7 (13.3) | 48.8 (16.6) | 39.0 (13.7) |
| weak, $m = 3$ | 570.1 (67.0) | 167.0 (13.5) | 100.0 (37.5) | 59.3 (23.4) |

ences between FS and BP-like are more pronounced when using a power friction law compared to a linear friction law (up to 17% compared to 8%, respectively). Furthermore, the weak implementation produces larger differences compared to the strong scheme. The P2P1 discretization reveals differences that are in between P1P1GLS-strong and P1P1GLS-weak results. Differences still tend to increase further below the highest employed resolution. Notably, the area-averaged differences are higher in the outlet region than in the ice stream region (see Fig. 5f).

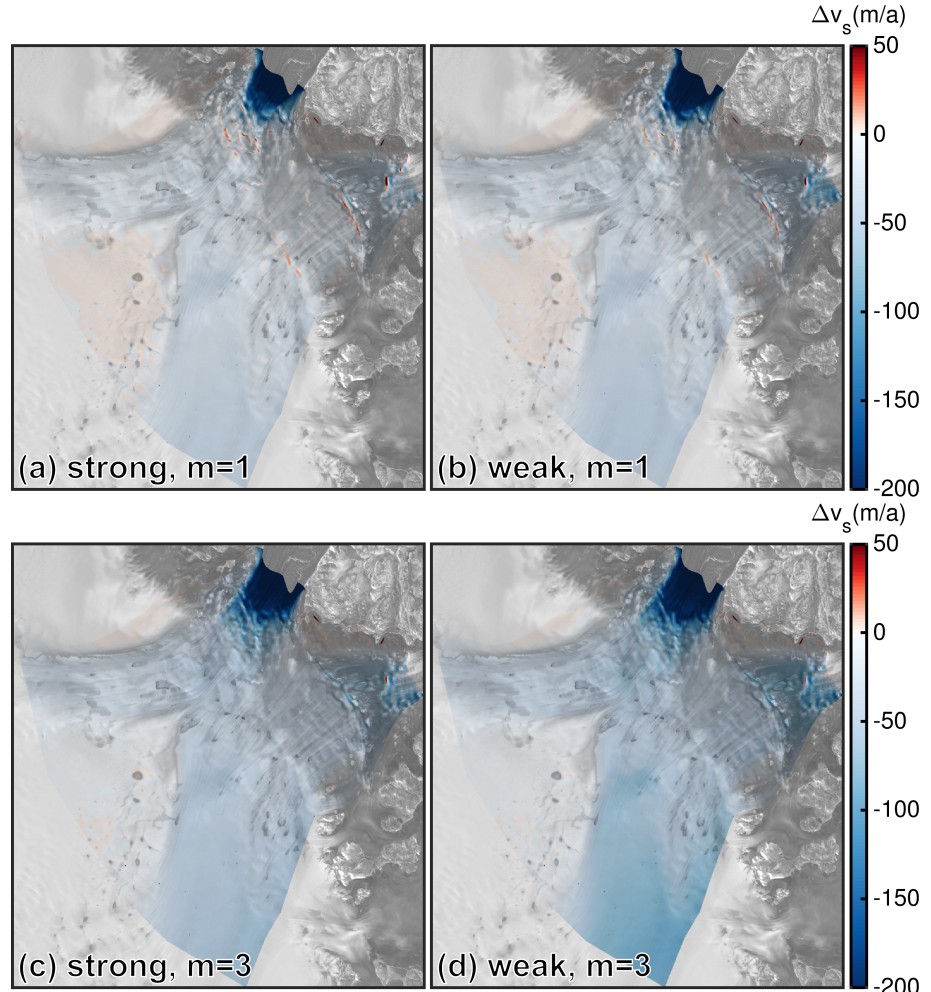

**Figure 7.** Difference of surface velocity $\Delta v_s^{\text{FS}-\text{BP-like}} = v_s^{\text{FS}} - v_s^{\text{BP-like}}$ in the outlet domain for a resolution of $150\,\text{m}$ and the P1P1GLS discretization. (a, b) Differences for $m = 1$ and the strong and weak friction implementation, respectively. (c, d) Differences for $m = 3$ and the strong and weak friction implementation, respectively. Please note the different color bar compared to Fig. 4. Background image is a RADARSAT Mosaic (Joughin, 2015; Joughin et al., 2016).

### 5.3 Impact on ice discharge

To estimate the impact on ice discharge, we calculated the ice mass flux along two flux gates of 79NG and ZI (see white lines in Fig. 2a). The flux gate of 79NG corresponds to the grounding line of the BM data set while the flux gate of ZI was shifted approx. 3 km upstream from the calving front. The ice mass flux is calculated as $Q = \int_\Gamma \bar{v} H \varrho_i ds$, where $\bar{v}$ indicates the depth-averaged ice velocity and the usage of the upper case $Q$ indicates integration over the flux gate area $\Gamma$ with the line segments $ds$. On average the total simulated flux for FS across 79NGs grounding line is $14.7\,\text{Gt}\,\text{a}^{-1}$ and $14.2\,\text{Gt}\,\text{a}^{-1}$ at the flux gate at

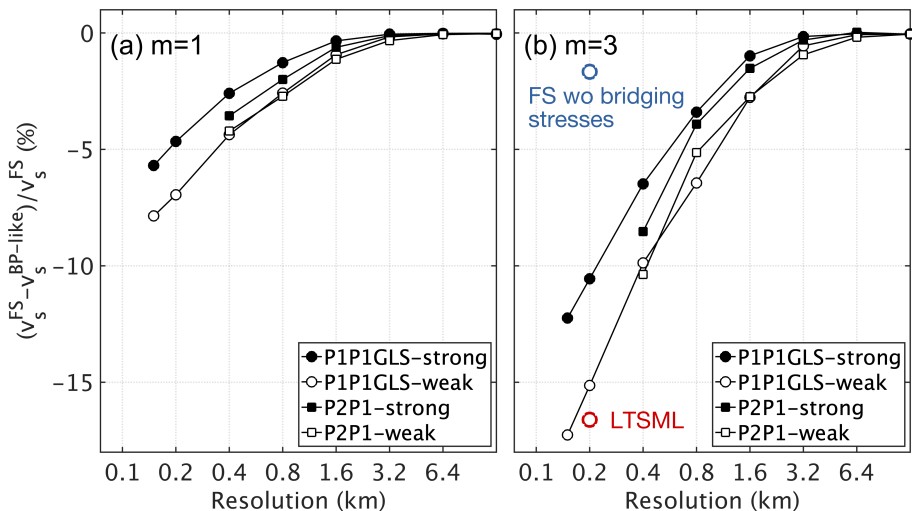

**Figure 8.** Relative surface velocity differences of area-averaged surface velocities between the FS and BP-like stress regimes in the outlet domain. (a) Linear friction law ($m = 1$). (b) Non linear friction law ($m = 3$). The blue and red circle show two simulations from a 'FS without (wo) bridging stresses' and LTSML-like regimes simulated with P1P1GLS-weak (see Discussion). For these two stress regimes, differences are calculated relative to FS.

ZIs terminus. Figure 9 displays relative ice mass flux differences between FS and BP-like. Overall, the BP-like model results
in higher discharge than FS, whereas the differences show a different magnitude: at 79NG, BP-like ice discharge is up to 40%
($m = 1$) and 50% ($m = 3$) higher than FS. In contrast, a moderate difference of about 5% in both friction laws is found for ZI.

### 5.4   Free surface evolution

In this section, we investigate how the velocity differences are represented in the solution of the ice surface position. The evolution of the free surfaces is given by the kinematic boundary condition

$$\frac{\partial z_s}{\partial t} = v_x n_x + v_y n_y + v_z n_z + \dot{a}_s, \tag{14}$$

where $\dot{a}_s(x, y)$ is the accumulation-ablation function at the surface. Here, we assume no melting or accumulation, i.e. $\dot{a}_s(x, y) = 0$, and the so-called emergence velocity ($\boldsymbol{v} \cdot \boldsymbol{n}$) determines the surface elevation change, $\partial z_s/\partial t$. Since we do not run a forward model the computed free surface evolution is representative for the first time step.

    Figure 10 shows the emergence velocity for FS and the difference between FS and BP-like. The FS emergence velocity
reveals a somewhat noisy pattern. Due to the hyperbolic nature of Eq. 14 numerical stabilization is necessary (Donea and Huerta, 2003; Riviere, 2008) for solving the free surface equation which certainly will smooths out sharp gradients and oscillations. However, the FS emergence velocity shows values in the range of $\pm 50 \, \mathrm{m \, a^{-1}}$ (higher at some isolated grid nodes), which is high but not very extreme. Considering values for surface slopes and velocities of the observational data sets (BM and MEaSURE) of up to 0.1 and $1000 \, \mathrm{m \, a^{-1}}$, respectively, lead to values of about $100 \, \mathrm{m \, a^{-1}}$ for the horizontal part of the

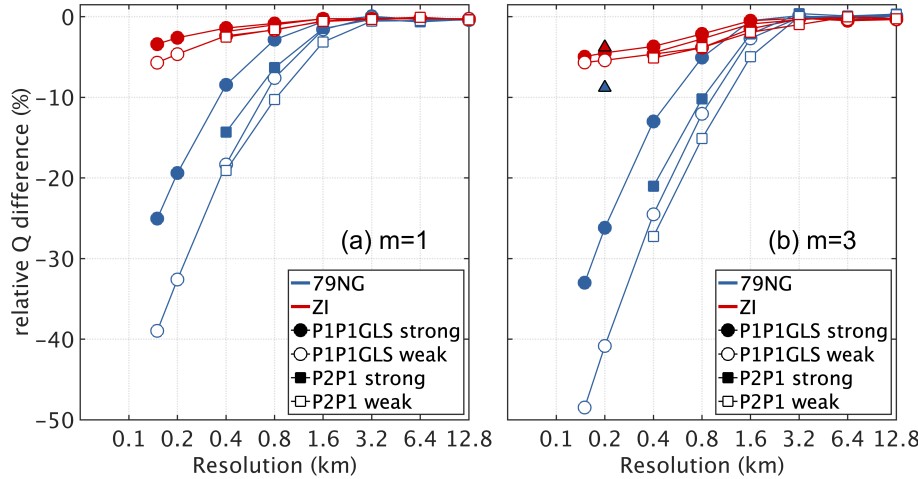

**Figure 9.** Relative ice mass flux differences between FS and the BP-like stress regimes ($\Delta Q^{\text{FS}-\text{BP-like}} = (Q^{\text{FS}} - Q^{\text{BP-like}})/Q^{\text{FS}}$). The location of the flux gates at 79NG and ZI is indicated in Fig. 2. On average the total simulated flux for FS at 79NG is 14.7 Gt a$^{-1}$ and 14.2 Gt a$^{-1}$ at ZI. The triangles in (b) indicate simulation using P1P1GLS-weak with a a resolution of $l = 200\,\text{m}$ but a re-gridded geometry from a simulation with $l = 1600\,\text{m}$ (see Sect. 6).

emergence velocity (i.e. $v_x n_x + v_y n_y$). Assuming an initial time step of 0.05 years the estimated maximum height change corresponds roughly to $\pm 2.5\,\text{m}$. Assuming ongoing elevation changes $\partial z_s / \partial t$ in that area of about 1 to $2\,\text{m a}^{-1}$ (derived from laser altimetry, pers. comm. Veit Helm, AWI, and older data published in Helm et al. (2014)) the estimated maximum ratio between emergence velocity and height changes ($\boldsymbol{v} \cdot \boldsymbol{n}/(\partial z_s / \partial t)$) is about 25. A transient relaxation run would smooth the surface slopes. However, the high values are not critical as we are interested in the differences between FS and BP-like.

The emergence velocity differences between FS and BP-like reveals a higher emergence velocity in FS, particularly at the grounding line at 79NG. According to Eq. 14, BP-like would therefore compute a surface elevation that is lower than the FS surface within the first time step. Although we are not running a time dependent simulation and, therefore, not capturing grounding line migration we evaluate the buoyancy balance from the stationary setup. The contact condition, i.e. $P_i > P_w$, at the ice base (e.g. Durand et al., 2009). BP-like reveals a grounding line that lies further upstream ($\approx 0.5\,\text{km}$) than the one calculated from FS. However both are still located within the hinge zone of 79NG.

## 5.5 Relevant physical processes

The reason for the difference in the simulated stress regime is a complex composition of ice dynamic processes and how they are resolved in the ice flow model, such as the basal boundary condition and the internal deformation. In addition, basal drag is coupled to internal deformation via the effective viscosity, making it difficult to separate the two entangled processes. Nevertheless, we strive to give a flavour of the origins of the model differences by analyzing key processes of ice dynamics below.

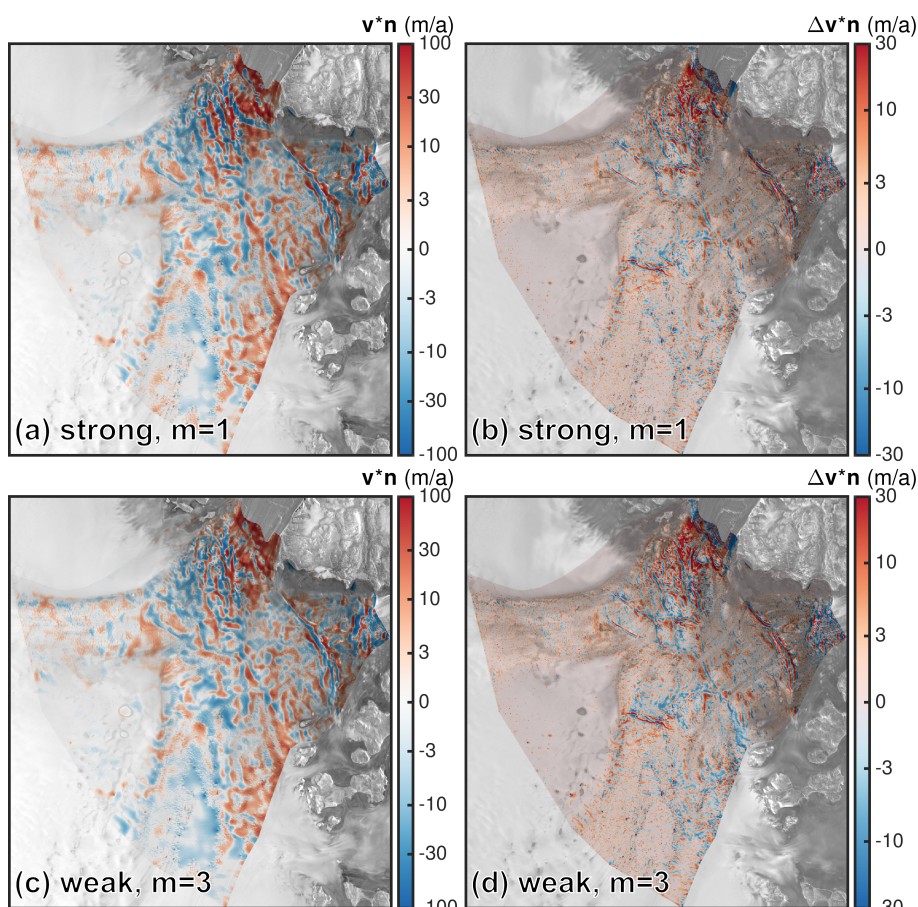

**Figure 10.** Evaluation of the kinematic boundary condition (emergence velocity $\boldsymbol{v} \cdot \boldsymbol{n}$) at the surface for $l = 150\,\text{m}$. Surface mass balance is assumed to be zero. Upper row shows (a) FS emergence velocity and (b) emergence velocity differences between FS and BP-like ($\boldsymbol{v} \cdot \boldsymbol{n}|_{\text{FS}} - \boldsymbol{v} \cdot \boldsymbol{n}|_{\text{BP-like}}$) for $m = 1$ and P1P1GLS-strong. Panels (c) and (d) show the equivalent fields for $m = 3$ and P1P1GLS-weak. Background image is a RADARSAT Mosaic (Joughin, 2015; Joughin et al., 2016).

Figure 11 and 12 show the difference of characteristic quantities between FS and BP-like for the ice-stream and outlet domain, respectively. First of all we would expect, that FS would lead to larger ice velocities than BP due to the effective deformation rate (Eq. 9). FS captures the entire stress tensor, which is equivalent to making ice softer on average for the same flow parameters (assuming that strain-rate components shared by FS and BP-like have the same magnitude). Indeed, the depth-averaged viscosity, $\bar{\eta}$, is basically lower for FS compared to BP-like (Fig. 11g–i and 12c,f). For $E = 0.1$ the depth-averaged viscosity is lower in some larger areas, while for $E = 6$ the pattern changes towards several smaller patches; the magnitude of $\bar{\eta}$ among the $E$ values changes by one order.

However, since FS generally reveals lower velocities than BP-like, the softer ice in FS appears to be compensated by another process. A fundamental control on ice flow is the basal drag indicating the glacier's behaviour for fast flow. The spatial differ-

ences of basal drag inferred from the two models is generally larger for FS compared to BP (Fig. 11d–f and 12b, e). The spatial patterns generally coincide with the differences in surface velocity (Fig. 4a and 7). In the ice stream region, the difference is largest for $E = 0.1$ (up to 140 kPa between FS and BP-like). In this setup, it seems that the basal drag is an essential control of the ice flow. With softer ice, the difference in the basal drag decreases. In the outlet domain, the basal drag difference is more pronounced for the weak implementation of the basal boundary condition with $m = 3$ compared to the strong implementation with $m = 1$. At the 79NG and ZI area, the maximum difference in basal drag is 316 kPa (99%) and 121 kPa (73%) for the weak implementation with $m = 3$; for the strong setup with $m = 1$ it is 227 kPa (72%) and 96 kPa (55%), respectively.

Inspired by the findings of Gudmundsson (2003) and Hindmarsh (2004), deduced from idealized setups, we explore whether the criteria when a BP solution is invalid (BP invalid in regions of high slip ratio, high aspect ratio and high topographic variations) hold in our realistic setup. In Fig. 13 and 14 we present two-dimensional histograms of the relative velocity difference between FS and BP-like for discrete bins of FS slip ratio $v_b/v_s$, the local aspect ratio, $\epsilon = H/l$, and the bed slope, $|\nabla h_b|$ (see detailed line plots Figs. S4-S7)). Simulated differences emerge with increasing aspect ratio, $\epsilon$. With increasing aspect ratio neglecting terms of $\mathcal{O}(\epsilon^2)$, i.e. $\partial v_z/\partial x$ and $\partial v_z/\partial y$, the BP-like (and also BP) model becomes problematic and the solution inaccurate. Similarly, the difference between FS and BP is most pronounced in region with a high slip ratio (i.e., near-plug-flow conditions prevail), while with a higher resolution a larger number of grid nodes are involved in producing this error. At a first glance, the distribution of the relative error with respect to the bed slope unveils larger errors at smaller slopes. However, the distribution is certainly biased by large errors occurring in the fast flowing and smooth trunk of the NEGIS region (see Fig. 2). By neglecting the bed slope bin from 0 to 0.05, the higher bed undulations lead to larger errors at higher resolutions; however, the bed slope seems to play a secondary role, compared to e.g. the slip ratio, as fewer grid nodes are involved. The analysis underlines that the source of stress regime deviations occurs in regions of high slip ratio, high aspect ratio and high topographic variability. A similar conclusion for mountain glaciers was drawn in Le Meur et al. (2004) by comparing FS and SIA.

## 6   Discussion

At the ice stream region, the quantitative comparison of both stress regimes reveals that, particularly in extremes cases ($E = 0.1$ and $E = 6$), the BP-like solution produces higher surface velocities than FS ($E = 0.1$) and a large spread around the FS solution ($E = 6$). However, these extreme cases at such a high resolution are to date rarely used in the community. For the $E = 1$ ('classical case') and $E = 3$, we found distinct velocity differences, but those are very low (about 1.5% in spatially averaged surface velocities). Although these velocity differences are small, they tend to increase further at the highest grid resolution used. We cannot rule out the possibility that at a higher resolution the disagreements would be more substantial. However, since FS is computationally expensive, it is satisfying to note that the differences between the approximated model and the exact solution are small in resolutions that are currently being run within the ice sheet modelling community. Therefore, it might be favourable to use a BP stress regime instead of FS for most ice stream modelling studies.

However, once the outlet regions of NEGIS are included (i.e., 79NG and ZI), the analysis shows that differences between FS and BP-like are large in the 79NG grounding zone. This finding is consistent with the work by Morlighem et al. (2010) at

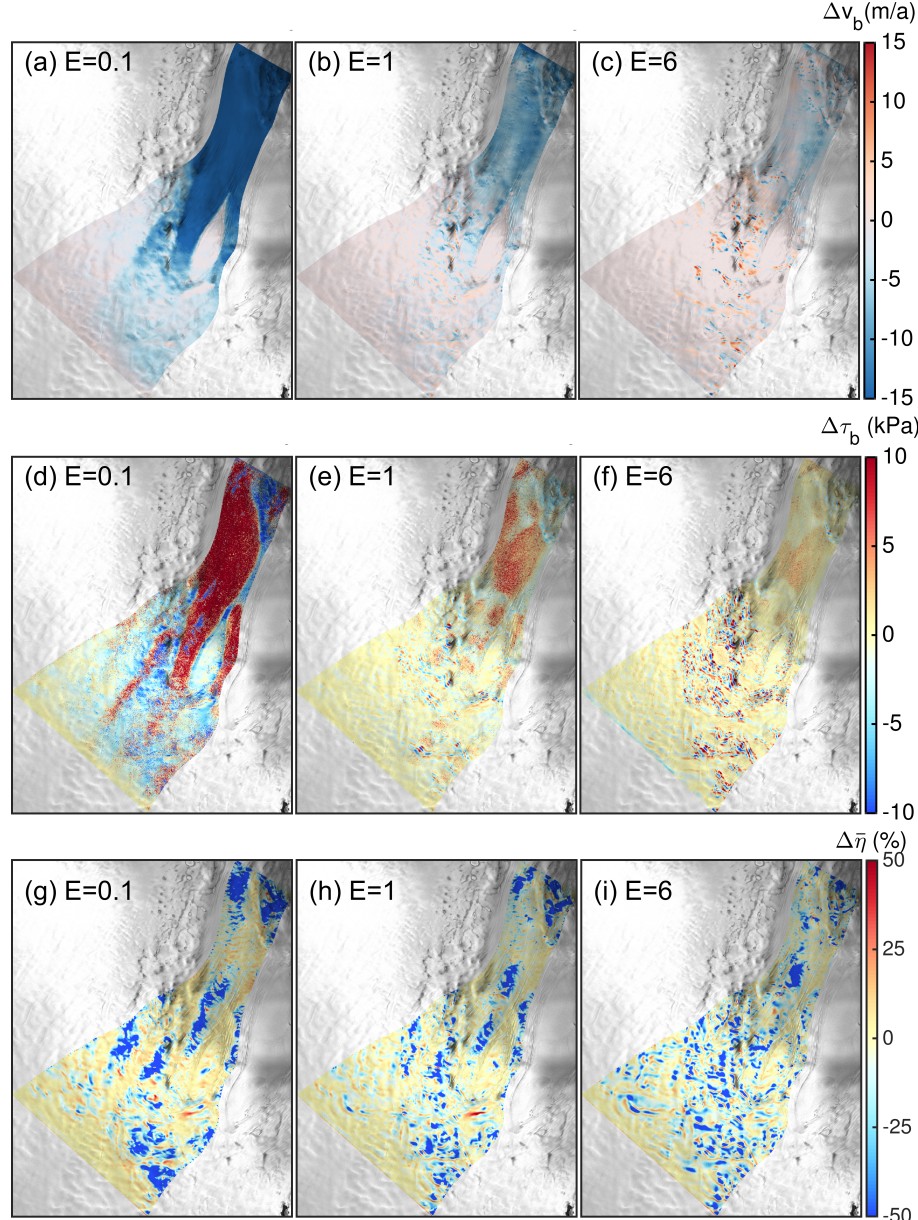

**Figure 11.** Comparison of characteristic quantities for a resolution of $l = 100\,\text{m}$, $m = 1$ and P1P1GLS-strong at the ice-stream domain. Difference (FS minus BP-like solution) of basal velocities, $v_b$ (a, b, c) and basal drag $\tau_b$ (d, e, f). Relative difference of depth-averaged viscosity $\bar{\eta}$ (g, h, i) between FS and BP-like. The domain averaged mean viscosity for FS and $E = 0.1$, 1, and 6 are 10.8, 3.6, and $1.1\times10^{14}\,\text{Pa}\,\text{s}$, respectively; for BP-like they are 12.6, 4.1, and $1.3\times10^{14}\,\text{Pa}\,\text{s}$, respectively. The equivalent fields for $E = 3$ reveal patterns and magnitudes that range from $E = 1$ to $E = 6$. Background image is a RADARSAT Mosaic (Joughin, 2015; Joughin et al., 2016).

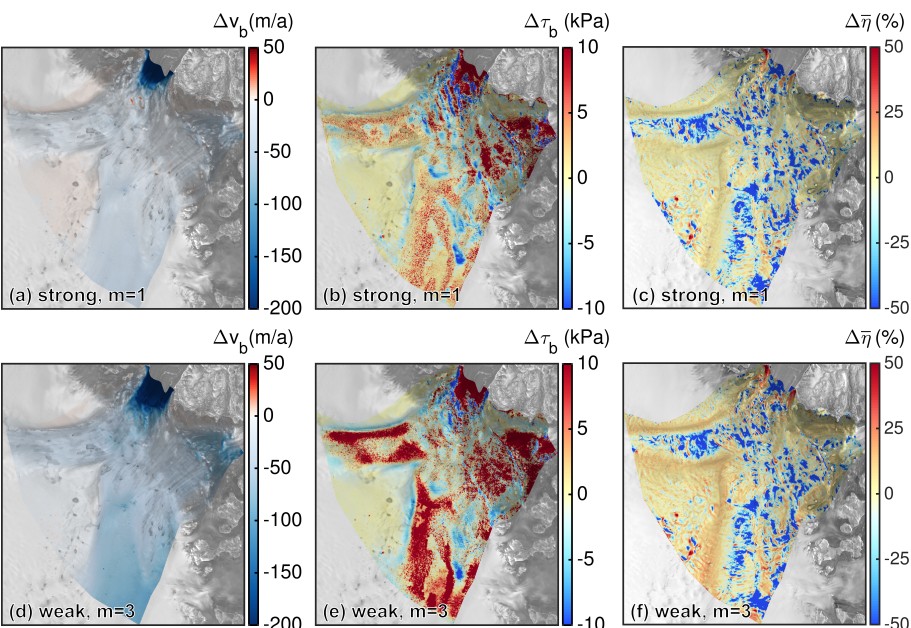

**Figure 12.** Comparison of characteristic quantities for a resolution of $l = 150\,\mathrm{m}$ and $E = 1$ at the outlet domain. Upper row (a, b, c) shows absolute difference of basal velocities, $v_b$ (a), basal drag $\tau_b$ (b), and relative difference of depth-averaged viscosity $\bar{\eta}$ (c) for $m = 1$ and P1P1GLS-strong. The domain averaged mean viscosities for FS are $3.00\times10^{14}\,\mathrm{Pa\,s}$ and $3.40\times10^{14}\,\mathrm{Pa\,s}$ for BP-like. Lower row (d, e, f) shows the equivalent fields but for $m = 3$ and P1P1GLS-weak. The domain averaged mean viscosity for FS is $3.11\times10^{14}\,\mathrm{Pa\,s}$ and $3.39\times10^{14}\,\mathrm{Pa\,s}$ for BP-like. Background image is a RADARSAT Mosaic (Joughin, 2015; Joughin et al., 2016).

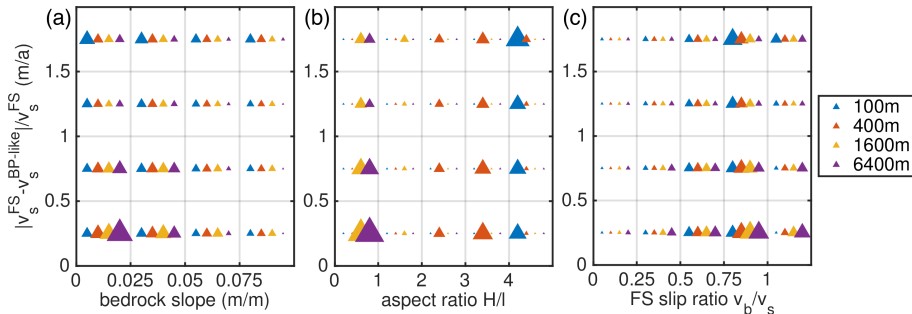

**Figure 13.** 2D dimensional histograms for $E = 1$, $m = 1$, P1P1GLS-strong and four different grid resolutions for the ice-stream region. Counts of grid nodes per relative velocity error bin and per (a) bed slope bin, (b) local aspect ratio bin, and (c) FS slip ratio bin. The sizes of the triangles are normalized by the total number of horizontal grid nodes of each mesh.

PIG. The overestimation of BP-like surface velocities to FS in our study agrees with their necessity of a lower inferred basal drag for FS to reproduce observed velocities compared to BP. They attribute this behavior to the developing bridging stresses in FS ($t_{zz}$ about 2% larger in FS compared to BP) due to the rising bed towards the grounding line, which in turn causes the

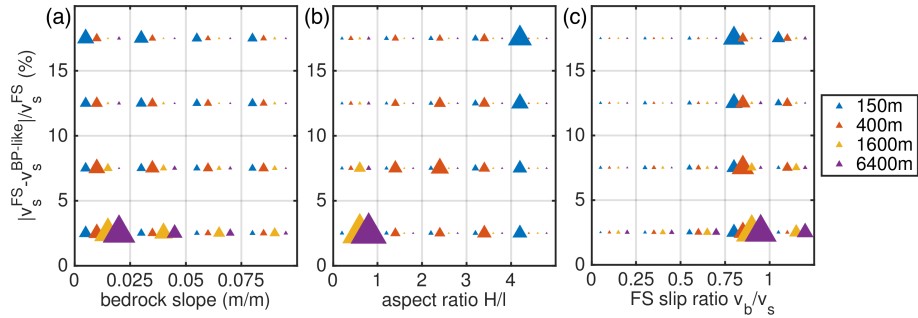

**Figure 14.** 2D dimensional histograms for $E = 1$, $m = 3$, P1P1GLS-weak and four different grid resolutions for the outlet region. Counts of grid nodes per relative velocity error bin and per (a) bed slope bin, (b) local aspect ratio bin, and (c) FS slip ratio bin. The sizes of the triangles are normalized by the total number of horizontal grid nodes of each mesh.

reduction in basal drag. However, in our simulations a clear connection to bridging stresses could not be identified. Figure 15 (panels a and c) shows the relative differences of $t_{zz}$ between FS and BP-like. Although the differences are in a similar range ($\pm 2\%$) as in Morlighem et al. (2010) there is no clear trend that could explain the lower velocities in FS. However, at PIG the
bed rises towards the grounding line, while at 79NG the bed mainly slopes down towards the grounding line. This geometric setting might control how the stresses develop. In our case, the differences mainly stem from the basal drag (Fig. 11d–f and 12b, e) that is for most areas higher in FS compared to BP-like. The regions where BP-like simulates a lower basal drag than FS overlap with regions where the assumption that horizontal gradients of the vertical velocity are small compared to the vertical gradient of the horizontal velocity is invalid (Fig. 15b, d). These terms are of similar order and therefore, dropping stress terms
of $\mathcal{O}(\epsilon^2)$, i.e. $\partial v_z / \partial x$ and $\partial v_z / \partial y$, lead to a lower basal drag in the BP-like case. Consequently, we assume that basal sliding is overestimated in BP-like simulations.

In order to highlight that bridging stresses play a minor role in our setup, we have run two other stress regime versions (executed with P1P1GLS-weak and up to resolution of $l = 400\,\mathrm{m}$). The first version keeps the bridging stresses in the third equation of the momentum balance (Eq. 8; this is similar to LTSML (van der Veen and Whillans, 1989; Kleiner and Humbert,
2014) according to Hindmarsh (2004)); the second version is a FS model without ('wo') bridging stresses. The area-averaged results reveal that LTSML-like is close to BP-like while 'FS wo bridging stresses' is rather similar to FS (Fig 8b).

In our diagnostic simulations, we compute lower ice mass fluxes with FS compared to BP-like. This supports the tendency that the sea level rise estimate of FS models is lower than the one of simpler models. (Seddik et al., 2012; Favier et al., 2014; Seddik et al., 2017). However, we observe a different response at 79NG and ZI. At 79NG, ice discharge differences are up
to 50%, while at ZI, ice discharge differences are about 5%. Also, at 79NG, differences seem to increase below the highest resolution employed here, while at ZI, the anticipated discharge difference increase appears to be moderate. A thorough analysis of why these two regions behave differently unveiled no significant differences between ice dynamic processes (e.g., basal drag shows a similar decrease). However, a potential source might be the geometric setting, e.g., the data coverage of available in-situ retrieved data leads to different topographic variability in both regions. To test whether the response behaviour is an

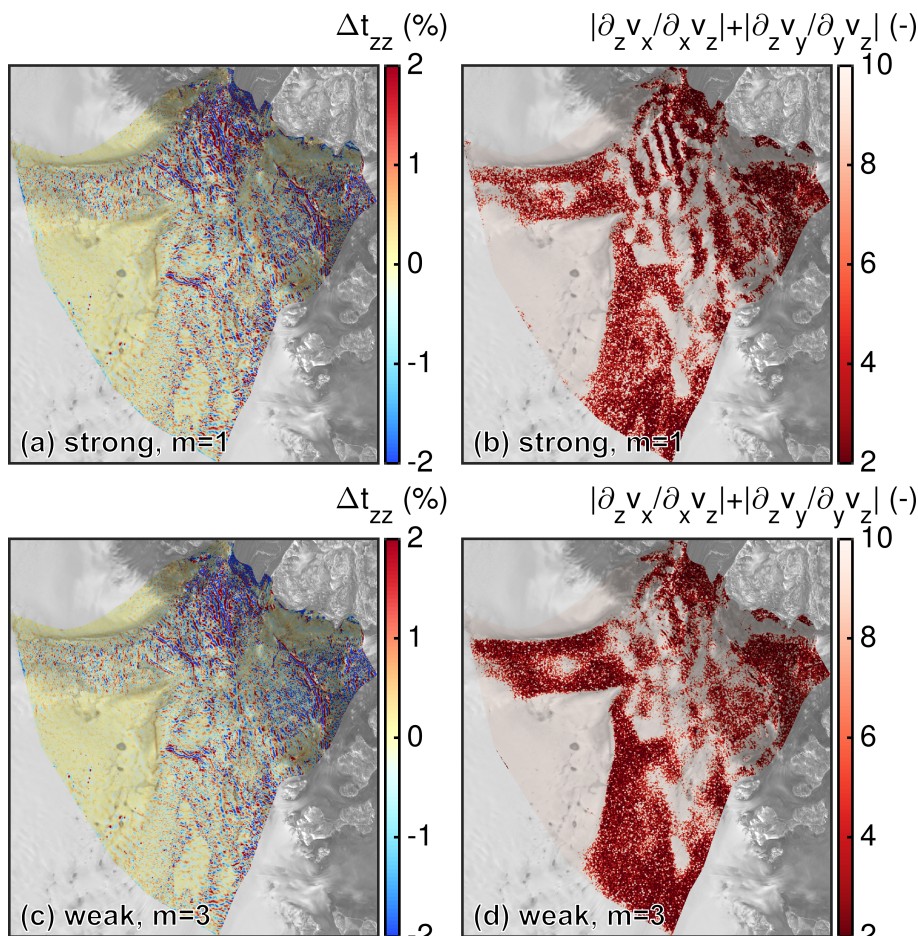

**Figure 15.** Comparison of (a, c) Relative difference of $t_{zz}$ at the ice base between FS and BP-like and (b, d) ratio of vertical gradient of the horizontal velocity to horizontal gradients of the vertical velocity ($|(\partial v_x/\partial z)/(\partial v_z/\partial x)| + |(\partial v_y/\partial z)/(\partial v_z/\partial y)|$) in absolute terms from the FS solution. Upper row (a, b) shows results for $l = 150\,\text{m}$, $E = 1$, $m = 1$ and P1P1GLS-strong. Lower row (c, d) shows results for $l = 150\,\text{m}$, $E = 1$, $m = 3$ and P1P1GLS-weak. Background image is a RADARSAT Mosaic (Joughin, 2015; Joughin et al., 2016).

effect of topographic variability, we re-run a simulation with P1P1GLS-weak, $m = 3$ and $l = 200\,\text{m}$ but using the geometry from a re-gridded $l = 1600\,\text{m}$ simulation. Re-gridding from a coarser geometry acts as smoothing and leads to a less variable geometry. The simulated discharge reveals that 79NG and ZI respond somewhat similarly (see triangles in Fig. 9b). However, whether the different response at 79NG and ZI with the high-resolution bedrock stems from in-situ data coverage or the fact that ZI has generally a smoother geometry than 79NG must be examined in further studies.

We tested the numerical robustness of the underlying finite element method by employing different discretization schemes and analysing how the non-penetration condition at the ice base is enforced. In general, the different schemes show a similar velocity difference pattern between FS and BP-like. For $m = 1$ and $m = 3$ the area averaged velocity differences for $l = 150\,\text{m}$

are within $-6.1\%$ to $-7.6\%$ and $-13.1\%$ to $-16.2\%$, respectively. Therefore, the choice of the employed friction law plays a larger role than the choice of numerical characteristics tested in this study.

However, some tests regarding the numerical robustness are missing. Because we keep the number of layers constant over the entire modelling domains, large variations in the aspect ratio occur when refining the mesh. The element aspect ratio is critical for the numerical error and the classical BP system would suffer less under bad element aspect ratios (Larour et al., 2012, Sect 5.1.3 therein). In order to assess the impact of element aspect ratios, we carried out a study where we varied the number of vertical layers from 3 to 11. For this test we re-run the simulations up to a resolution of $l = 400$ m with the P1P1GLS-strong

discretization scheme and $m = 3$ applied to the outlet domain. The area-averaged surface velocities difference between FS and BP-like are very small (Fig. S8). This demonstrates that the comparison is not suffering under bad element aspect ratios and the numerical error behave similar in FS and BP-like.

The basal boundary conditions are retrieved by inverting for the basal friction coefficient with BP employing a resolution of $l = 800$ m. That means, that the boundary conditions are not consistent with FS or with other resolutions. In order to study

whether this inconsistency influences the results, we repeat a set of experiments with two different fields for the friction coefficient. To this end, we run two ISSM friction inversions with (1) $E = 1$, $l = 200$ m, $m = 3$ and BP and (2) $E = 1$, $l = 800$ m, $m = 3$ and FS and feed the inferred friction fields to COMSOL. The area-averaged differences between FS and BP-like of the three setups show very minor differences (Fig. S9).

By implementing FS and BP-like solvers using the COMSOL finite element package, we have developed a useful tool for

exploring differences between FS and BP-like, without complications due to differing numerics. The developed model is also very flexible in order to test various ice flow components (e.g., by simply including bridging stresses to convert the BP-like model to LTSML-like). In our model setup, we solve for the same physical problem as in the 'classical' BP scheme but without taking advantage of the reduced number of equations to be solved. Regarding the model physics, our results compare well with other BP implementations, but our BP-like model is more expensive. Therefore, the developed BP-like model is not intended

for long time integrations or large ensembles.

One limitation of our study is that the simulations are not prognostic, i.e. we have not investigated how the solution differences propagate to or interact with other components of an ice sheet model, e.g. by coupling to the ice thickness evolution. The calculation of ice discharge and the emergence velocity demonstrates that the stress regimes show distinct discrepancies in a key process, but whether these differences are accumulated in the glaciers mass balance over transient runs, eventually leading

to substantially different estimates of ice mass changes, or fading out must be investigated in future studies.

A process not considered here is the thermo-mechanical coupling. For instance, within a thermo-mechanical model, the feedback of temperature on the ice viscosity, in turn, changes the ice flow. Additionally, there are several feasible processes how the stress regime differences can modify the thermal regime: (1) the different representation of the basal drag presumably generates a different amount of frictional heating, (2) the different vertical deformation produces a different internal heat source

rate, and (3) the internal advection scheme leads to different heat transport. Though the role of these feedbacks is not explored here, there are indications that small initial differences become much larger over long time integrations (Zhang et al., 2015).

## 7 Conclusions

We compared two approaches to represent ice flow dynamics, namely the FS and the BP-like formulation, for two different regions of the NEGIS. Both stress regimes are implemented within the same dynamic ice flow model to enable a consistent comparison where numerical differences are largely eliminated. Our comparison experiments between FS and BP-like stress regimes unveil that BP-like tends to overestimate surface velocities and the resulting ice discharge. The models disagreements start at a horizontal resolution of around 1000 m and continuously increase with finer horizontal resolution (down to 100 m).

In the ice stream region, when considering a rheology with much softer ice ($E = 6$) an enhanced spread in surface velocity differences between FS and BP-like was modelled. In contrast, stiffer ($E = 0.1$) ice leads to enhanced ice flow velocities in BP-like compared to FS. The classical rheology case ($E = 1$) reveals very moderate differences between FS and BP-like. However, once the outlet regions of NEGIS are included, i.e. 79NG and ZI, ice flow treatment with FS is essential important surface velocity differences occur (up to $570 \, \mathrm{m \, a^{-1}}$ at the grounding line of 79NG). Numerical characteristics play a minor role compared to the choice of the friction type: model differences are particularly important when a power-law friction is applied (as opposed to a linear approach). The basal drag and topographic undulations are identified as the general causes for the model disagreements.

Whether FS will reduce uncertainties of future sea-level projections must be examined further in subsequent transient studies. Although computer power is increasing, transient simulations in FS with a horizontal resolution of about 100 m in dynamic, relevant regions are still challenging. Our diagnostic simulations and previous studies indicate that FS and BP differ significantly in regions with a grounding zone, and results from non-FS models should therefore be viewed with caution.

*Code and data availability.* COMSOL Multiphysics© is a commercial software that is not freely available. The models used here are accessible for COMSOL users upon request to the authors. The ice flow model ISSM is open source and freely available at https://issm.jpl.nasa.gov/ (last access: April 6, 2022; Larour et al. (2012). Simulations results are available on Zenodo with digital object identifier: https://doi.org/10.5281/zenodo.64(
(Rückamp et al., 2022).

.

## Appendix A: Inversion with ISSM

The simulations presented make use of a basal friction coefficient, $k^2$, that is inferred by an inversion method. For the inversion of the basal friction coefficient, we operate the Ice-Sheet and Sea-level System Model (ISSM, Morlighem et al., 2010; Larour et al., 2012), an open source finite element flow model appropriate for continental-scale and outlet glacier applications. The setup is as described above, i.e. (1) domain outline and geometry data set is similar as in the COMSOL simulations (see Fig. 2), (2) ice rheology is taken from SICOPOLIS spinup (Rückamp et al., 2019) and interpolated onto the computational mesh and (3) the friction law follows the form in Eq. 6. But for the ease of computational time model calculations are performed on

a structured finite element grid with a horizontal resolution of $1\,$km and the BP-stress regime. The inversion is conducted separately for each enhancement factor $E = 0.1, 1, 3$ and $6$ and the sliding exponents $m = 1$ and $m = 3$.

Within the inverse problem a cost function ($J$), that measures the misfit between observed, $\mathbf{v}^{\mathrm{obs}} = (v_x^{\mathrm{obs}}, v_y^{\mathrm{obs}})$, and simulated velocities, $\mathbf{v} = (v_{\mathrm{y}}, v_{\mathrm{y}})$, is minimized. We use the observed velocities from the MEaSUREs project (Joughin et al., 2016, 2018) as target within the inversion. The cost function is composed of two terms which fit the velocities in fast- and slow-moving areas. A third term is a Tikhonov regularization to avoid oscillations due to over fitting. The cost function is defined as follows:

$$
J(\mathbf{v}, k) = \gamma_1 \frac{1}{2} \int_{\Gamma_s} (v_x - v_y^{\mathrm{obs}})^2 + v_y - v_y^{\mathrm{obs}})^2 d\Gamma_s + \gamma_2 \frac{1}{2} \int_{\Gamma_s} \ln \left( \frac{\sqrt{v_x^2 + v_y^2} + \varepsilon}{\sqrt{v_x^{\mathrm{obs}\,2} + v_y^{\mathrm{obs}\,2}} + \varepsilon} \right)^2 d\Gamma_s + \gamma_t \frac{1}{2} \int_{\Gamma_b} \nabla k \cdot \nabla k \, d\Gamma_b, \qquad \text{(A1)}
$$

where $\varepsilon$ is a minimum velocity used to avoid singularities and $\Gamma_s$ and $\Gamma_b$ are the ice surface and ice base, respectively. An L-curve analysis was performed to pick the Tikhonov parameter $\gamma_t$ (not shown). We obtained a good agreement to the observed velocities by choosing with $\gamma_1 = 1 \times 10^3$, $\gamma_2 = 1 \times 10^4$ and $\gamma_t = 1 \times 10^{-4}$ (see Fig. A1 and A2).

*Author contributions.* MR setup the COMSOL FS and BP model and the ISSM model, conducted the simulation and wrote large parts of the manuscript. AH designed the study. MR, TK and AH analyzed the results. All author contributed in writing the manuscript.

*Competing interests.* The authors declare that they have no conflict of interest.

*Acknowledgements.* We would like to thank the editor Harry Zekollari, an anonymous reviewer and Josefin Ahlkrona for the detailed reviews and constructive criticism that helped to improve the paper. The simulations were performed at AWI's cluster Cray-CS400 and at the North-German Supercomputing Alliance (HLRN). We would like to thank Natalja Rakowsky (AWI) for providing excellent support for COMSOL at computing facilities at AWI and HLRN. Ralf Greve (ILTS) kindly provided results from spin-up simulations with SICOPOLIS. We thank Eberhard Bänsch and Luca Wester for discussions on Stokes simulations (both Friedrich-Alexander-University of Erlangen).

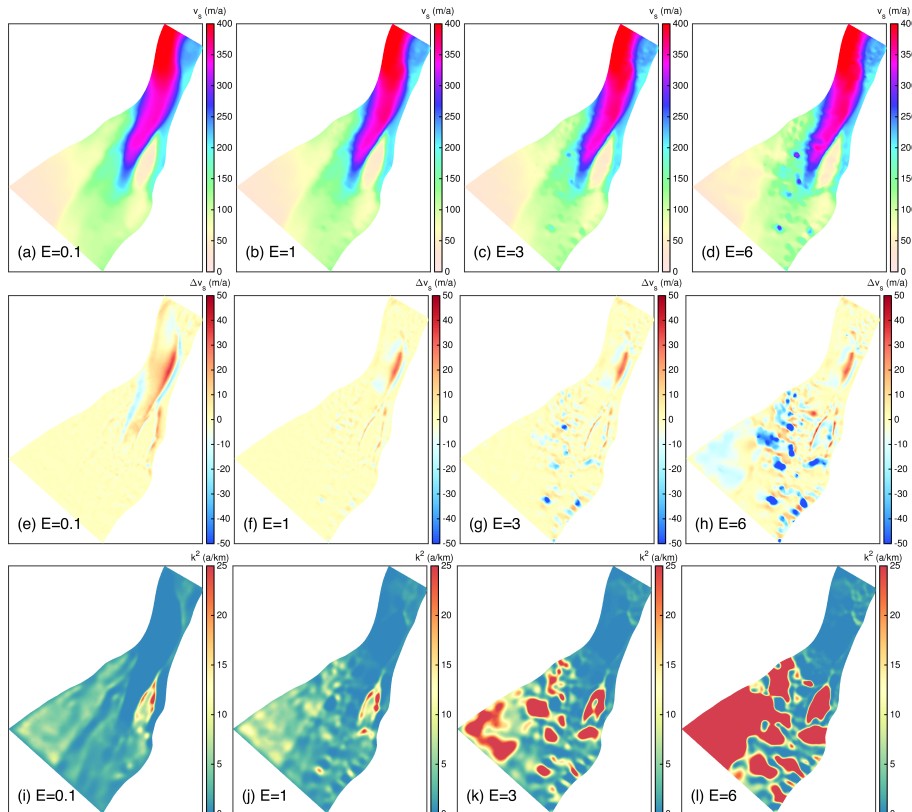

**Figure A1.** Overview of the ISSM inversion performed with the different values for $E = 0.1$, 1, 3 and 6 at the ice stream region. (a, b, c, d) Simulated surface velocities. (e, f, g, h). Velocity difference between observed and simulated velocities. (i, j, k, l) Inferred basal friction coefficient $k^2$.

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

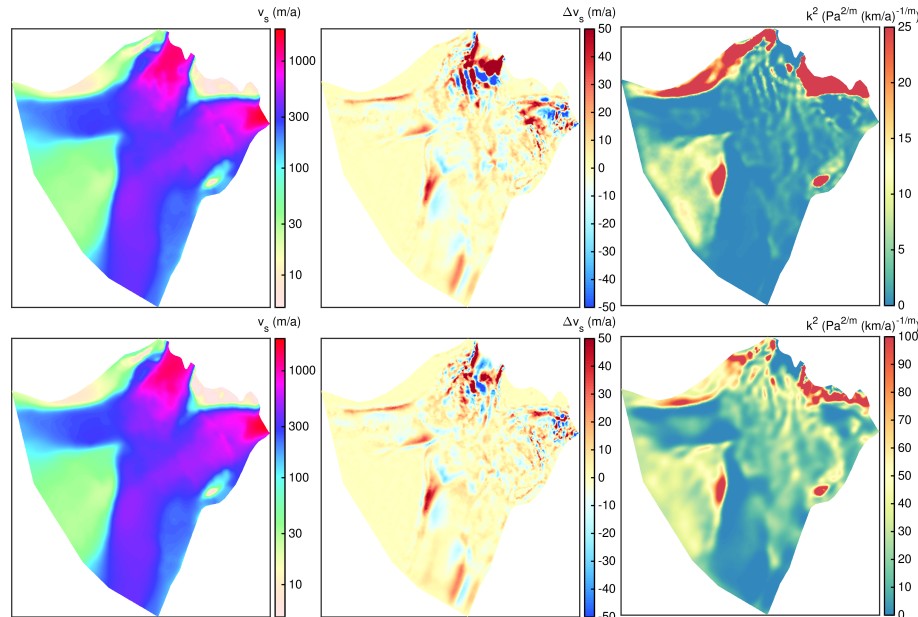

**Figure A2.** Overview of the ISSM inversion performed with the different values for $m = 1$ and $m = 3$ at the outlet region. Upper row shows simulated surface velocities (a), velocity difference between observed and simulated velocities (b), and the inferred basal friction coefficient $k^2$ (c) for $m = 1$. Lower row shows the equivalent fields for $m = 3$.

Budd, W. F., Keage, P. L., and Blundy, N. A.: Empirical Studies of Ice Sliding, Journal of Glaciology, 23, 157–170, https://doi.org/10.3189/S0022143000029804, 1979.

Bueler, E. and Brown, J.: Shallow shelf approximation as a "sliding law" in a thermodynamically coupled ice sheet model, J. Geophys. Res., 540     114, https://doi.org/10.1029/2008JF001179, 2009.

Cornford, S. L., Seroussi, H., Asay-Davis, X. S., Gudmundsson, G. H., Arthern, R., Borstad, C., Christmann, J., Dias dos Santos, T., Feldmann, J., Goldberg, D., Hoffman, M. J., Humbert, A., Kleiner, T., Leguy, G., Lipscomb, W. H., Merino, N., Durand, G., Morlighem, M., Pollard, D., Rückamp, M., Williams, C. R., and Yu, H.: Results of the third Marine Ice Sheet Model Intercomparison Project (MISMIP+), The Cryosphere, 14, 2283–2301, https://doi.org/10.5194/tc-14-2283-2020, 2020.

Cuffey, K. M. and Paterson, W. S. B.: The Physics of Glaciers, Elsevier, Amsterdam, The Netherlands etc., 4th edn., 2010.

de Boer, B., Dolan, A. M., Bernales, J., Gasson, E., Goelzer, H., Golledge, N. R., Sutter, J., Huybrechts, P., Lohmann, G., Rogozhina, I., Abe-Ouchi, A., Saito, F., and van de Wal, R. S. W.: Simulating the Antarctic ice sheet in the late-Pliocene warm period: PLISMIP-ANT, an ice-sheet model intercomparison project, The Cryosphere, 9, 881–903, https://doi.org/10.5194/tc-9-881-2015, 2015.

Donea, J. and Huerta, A.: Finite element methods for flow problems., Finite element methods for flow problems., 2003.

Durand, G., Gagliardini, O., de Fleurian, B., Zwinger, T., and Le Meur, E.: Marine ice sheet dynamics: Hysteresis and neutral equilibrium, J. Geophys. Res., 114, F03 009–, https://doi.org/10.1029/2008JF001170, 2009.

Favier, L., Durand, G., Cornford, S. L., Gudmundsson, G. H., Gagliardini, O., Gillet-Chaulet, F., Zwinger, T., Payne, A., and Le Brocq, A. M.: Retreat of Pine Island Glacier controlled by marine ice-sheet instability, Nature Climate Change, 4, 117–121, https://doi.org/10.1038/NCLIMATE2094, 2014.

Fowler, A. C. and Larson, D. A.: On the flow of polythermal glaciers - I. Model and preliminary analysis, R. Soc. Lond., p. 217–242, https://doi.org/10.1098/rspa.1978.0165, 1978.

Glen, J. W.: The Creep of Polycrystalline Ice, Proceedings of the Royal Society of London. Series A, Mathematical and Physical Sciences, 228, 519–538, https://doi.org/10.1098/rspa.1955.0066, 1955.

Goelzer, H., Nowicki, S., Edwards, T., Beckley, M., Abe-Ouchi, A., Aschwanden, A., Calov, R., Gagliardini, O., Gillet-Chaulet, F., Golledge,
N. R., Gregory, J., Greve, R., Humbert, A., Huybrechts, P., Kennedy, J. H., Larour, E., Lipscomb, W. H., Le clec'h, S., Lee, V., Morlighem, M., Pattyn, F., Payne, A. J., Rodehacke, C., Rückamp, M., Saito, F., Schlegel, N., Seroussi, H., Shepherd, A., Sun, S., van de Wal, R., and Ziemen, F. A.: Design and results of the ice sheet model initialisation experiments initMIP-Greenland: an ISMIP6 intercomparison, The Cryosphere, 12, 1433–1460, https://doi.org/10.5194/tc-12-1433-2018, 2018.

Goelzer, H., Nowicki, S., Payne, A., Larour, E., Seroussi, H., Lipscomb, W. H., Gregory, J., Abe-Ouchi, A., Shepherd, A., Simon, E., Agosta,
C., Alexander, P., Aschwanden, A., Barthel, A., Calov, R., Chambers, C., Choi, Y., Cuzzone, J., Dumas, C., Edwards, T., Felikson, D., Fettweis, X., Golledge, N. R., Greve, R., Humbert, A., Huybrechts, P., Le clec'h, S., Lee, V., Leguy, G., Little, C., Lowry, D. P., Morlighem, M., Nias, I., Quiquet, A., Rückamp, M., Schlegel, N.-J., Slater, D. A., Smith, R. S., Straneo, F., Tarasov, L., van de Wal, R., and van den Broeke, M.: The future sea-level contribution of the Greenland ice sheet: a multi-model ensemble study of ISMIP6, The Cryosphere, 14, 3071–3096, https://doi.org/10.5194/tc-14-3071-2020, 2020.

Gudmundsson, G. H.: Transmission of basal variability to a glacier surface, Journal of Geophysical Research: Solid Earth, 108, 2253, https://doi.org/10.1029/2002JB002107, 2003.

Hauke, G. and Hughes, T.: A unified approach to compressible and incompressible flows, Computer Methods in Applied Mechanics and Engineering, 113, 389–395, https://doi.org/10.1016/0045-7825(94)90055-8, 1994.

Helanow, C. and Ahlkrona, J.: Stabilized equal low-order finite elements in ice sheet modeling – accuracy and robustness, Computational
Geosciences, 22, 951–974, https://doi.org/10.1007/s10596-017-9713-5, 2018.

Helm, V., Humbert, A., and Miller, H.: Elevation and elevation change of Greenland and Antarctica derived from CryoSat-2, The Cryosphere, 8, 1539–1559, https://doi.org/10.5194/tc-8-1539-2014, 2014.

Hindmarsh, R. C. A.: A numerical comparison of approximations to the Stokes equations used in ice sheet and glacier modeling, Journal of Geophysical Research: Earth Surface, 109, https://doi.org/10.1029/2003JF000065, 2004.

Howat, I. M., Negrete, A., and Smith, B. E.: The Greenland Ice Mapping Project (GIMP) land classification and surface elevation data sets, The Cryosphere, 8, 1509–1518, https://doi.org/10.5194/tc-8-1509-2014, 2014.

Hutter, K.: Theoretical glaciology : material science of ice and the mechanics of glaciers and ice sheets, Reidel/Terra Pub. Co., Dordrecht, 1983.

Huybrechts, P.: The Antarctic ice sheet and environmental change: a three-dimensional modelling study = Der antarktische Eisschild und
globale Umweltveränderungen : eine dreidimensionale Modellstudie, https://doi.org/10.2312/BzP_0099_1992, 1992.

IPCC: Climate Change 2013: The Physical Science Basis. Contribution of Working Group I to the Fifth Assessment Report of the Intergovernmental Panel on Climate Change, Cambridge University Press, Cambridge, United Kingdom and New York, NY, USA, https://doi.org/10.1017/CBO9781107415324, 2013.

John, V. and Matthies, G.: Higher-order finite element discretizations in a benchmark problem for incompressible flows, International Journal
for Numerical Methods in Fluids, 37, 885–903, https://doi.org/10.1002/fld.195, 2001.

Joughin, I.: MEaSUREs Greenland Ice Sheet Mosaics from SAR Data, Version 1. Boulder, Colorado USA. NASA National Snow and Ice Data Center Distributed Active Archive Center, https://doi.org/10.5067/6187DQUL3FR5, 2015.

Joughin, I., Smith, B. E., Howat, I. M., Moon, T., and Scambos, T. A.: A SAR record of early 21st century change in Greenland, Journal of Glaciology, 62, 62–71, https://doi.org/10.1017/jog.2016.10, 2016.

Joughin, I., Smith, B. E., and Howat, I. M.: A complete map of Greenland ice velocity derived from satellite data collected over 20 years, Journal of Glaciology, 64, 1–11, https://doi.org/10.1017/jog.2017.73, 2018.

Kleiner, T. and Humbert, A.: Numerical simulations of major ice streams in western Dronning Maud Land, Antarctica, under wet and dry basal conditions, Journal of Glaciology, 60, 215–232, https://doi.org/doi:10.3189/2014JoG13J006, 2014.

Larour, E., Seroussi, H., Morlighem, M., and Rignot, E.: Continental scale, high order, high spatial resolution, ice sheet modeling using the
Ice Sheet System Model (ISSM), Journal of Geophysical Research, 117, F01 022, https://doi.org/10.1029/2011JF002140, 2012.

Le Meur, E., Gagliardini, O., Zwinger, T., and Ruokolainen, J.: Glacier flow modelling: a comparison of the Shallow Ice Approximation and the full-Stokes solution, Comptes Rendus Physique, 5, 709–722, https://doi.org/10.1016/j.crhy.2004.10.001, ice: from dislocations to icy satellites, 2004.

Leysinger Vieli, G. J.-M. C. and Gudmundsson, G. H.: On estimating length fluctuations of glaciers caused by changes in climatic forcing,
Journal of Geophysical Research: Earth Surface, 109, https://doi.org/10.1029/2003JF000027, 2004.

Lliboutry, L. and Duval, P.: Various isotropic and anisotropic ices found in glaciers and polar ice caps and their corresponding rheologies, Annales Geophysicae, 3, 207–224, 1985.

Ma, Y., Gagliardini, O., Ritz, C., Gillet-Chaulet, F., Durand, G., and Montagnat, M.: Enhancement factors for grounded ice and ice shelves inferred from an anisotropic ice-flow model, Journal of Glaciology, 56, 805–812, https://doi.org/10.3189/002214310794457209, 2010.

Meredith, M., Sommerkorn, M., Cassotta, S., Derksen, C., Ekaykin, A., Hollowed, A., Kofinas, G., Mackintosh, A., Melbourne-Thomas, J., Muelbert, M., Ottersen, G., Pritchard, H., and Schuur, E.: Polar Regions, in: IPCC Special Report on the Ocean and Cryosphere in a Changing Climate, edited by Pörtner, H.-O., Roberts, D., Masson-Delmotte, V., Zhai, P., Tignor, M., Poloczanska, E., Mintenbeck, K., Alegría, A., Nicolai, M., Okem, A., Petzold, J., Rama, B., and Weyer, N., chap. 3, pp. 203–320, Cambridge University Press, Cambridge, United Kingdom and New York, NY, USA, https://doi.org/10.1017/9781009157964.005, 2019.

Morlighem, M., Rignot, E., Seroussi, H., Larour, E., Dhia, H., and Aubry, D.: Spatial patterns of basal drag inferred using control methods from a full-Stokes and simpler models for Pine Island Glacier, West Antarctica, Geophysical Research Letters, 37, L14 502, 2010.

Morlighem, M., Williams, C. N., Rignot, E., An, L., Arndt, J. E., Bamber, J. L., Catania, G., Chauché, N., Dowdeswell, J. A., Dorschel, B., Fenty, I., Hogan, K., Howat, I., Hubbard, A., Jakobsson, M., Jordan, T. M., Kjeldsen, K. K., Millan, R., Mayer, L., Mouginot, J., Noël, B. P. Y., O'Cofaigh, C., Palmer, S., Rysgaard, S., Seroussi, H., Siegert, M. J., Slabon, P., Straneo, F., van den Broeke,
M. R., Weinrebe, W., Wood, M., and Zinglersen, K. B.: BedMachine v3: Complete bed topography and ocean bathymetry mapping of Greenland from multibeam echo sounding combined with mass conservation, Geophysical Research Letters, 44, 11 051–11 061, https://doi.org/10.1002/2017GL074954, 2017.

Nowicki, S. M. J., Payne, A., Larour, E., Seroussi, H., Goelzer, H., Lipscomb, W., Gregory, J., Abe-Ouchi, A., and Shepherd, A.: Ice Sheet Model Intercomparison Project (ISMIP6) contribution to CMIP6, Geoscientific Model Development, 9, 4521–4545,
https://doi.org/10.5194/gmd-9-4521-2016, 2016.

Oppenheimer, M., Glavovic, B., Hinkel, J., van de Wal, R., Magnan, A., Abd-Elgawad, A., Cai, R., Cifuentes-Jara, M., DeConto, R., Ghosh, T., Hay, J., Isla, F., Marzeion, B., Meyssignac, B., and Sebesvari, T.: Sea level rise and implications for low-lying islands, coasts and communities, in: IPCC Special Report on the Ocean and Cryosphere in a Changing Climate, edited by Pörtner, H.-O., Roberts, D., Masson-Delmotte, V., Zhai, P., Tignor, M., Poloczanska, E., Mintenbeck, K., Alegría, A., Nicolai, M., Okem, A., Petzold,

J., Rama, B., and Weyer, N., chap. 4, pp. 321–445, Cambridge University Press, Cambridge, United Kingdom and New York, NY, USA, https://doi.org/10.1017/9781009157964.006, 2019.

Pattyn, F.: Ice-sheet modelling at different spatial resolutions: focus on the grounding zone, Annals of Glaciology, 31, 211–216, https://doi.org/10.3189/172756400781820435, 2000.

Pattyn, F.: A new three-dimensional higher-order thermomechanical ice-sheet model: basic sensitivity, ice-stream development and ice flow
across subglacial lakes, Journal of Geophysical Research, 108, 2382, https://doi.org/10.1029/2002JB002329, 2003.

Pattyn, F., Perichon, L., Aschwanden, A., Breuer, B., de Smedt, B., Gagliardini, O., Gudmundsson, G. H., Hindmarsh, R. C. A., Hubbard, A., Johnson, J. V., Kleiner, T., Konovalov, Y., Martin, C., Payne, A. J., Pollard, D., Price, S., Rückamp, M., Saito, F., Souček, O., Sugiyama, S., and Zwinger, T.: Benchmark experiments for higher-order and full-Stokes ice sheet models (ISMIP-HOM), The Cryosphere, 2, 95–108, https://doi.org/10.5194/tc-2-95-2008, 2008.

Pattyn, F., Schoof, C., Perichon, L., Hindmarsh, R. C. A., Bueler, E., de Fleurian, B., Durand, G., Gagliardini, O., Gladstone, R., Goldberg, D., Gudmundsson, G. H., Huybrechts, P., Lee, V., Nick, F. M., Payne, A. J., Pollard, D., Rybak, O., Saito, F., and Vieli, A.: Results of the Marine Ice Sheet Model Intercomparison Project, MISMIP, The Cryosphere, 6, 573–588, https://doi.org/10.5194/tc-6-573-2012, 2012.

Pattyn, F., Perichon, L., Durand, G., Favier, L., Gagliardini, O., Hindmarsh, R., Zwinger, T., Albrecht, T., Cornford, S., Docquier, D., Fürst, J. J., Goldberg, D., Gudmundsson, G. H., Humbert, A., Hütten, M., Huybrechts, P., Jouvet, G., Kleiner, T., Larour, E., Martin,
D., Morlighem, M., Payne, A. J., Pollard, D., Rückamp, M., Rybak, O., Seroussi, H., Thoma, M., and Wilkens, N.: Grounding-line migration in plan-view marine ice-sheet models: results of the ice2sea MISMIP3d intercomparison, Journal of Glaciology, 59, 410–422, https://doi.org/doi:10.3189/2013JoG12J129, 2013.

Riviere, B.: Discontinuous Galerkin methods for solving elliptic and parabolic equations, Society for Industrial and Applied Mathematics, https://doi.org/10.1137/1.9780898717440, 2008.

Rückamp, M., Greve, R., and Humbert, A.: Comparative simulations of the evolution of the Greenland ice sheet under simplified Paris Agreement scenarios with the models SICOPOLIS and ISSM, Polar Science, 21, 14–25, https://doi.org/10.1016/j.polar.2018.12.003, 2019.

Rückamp, M., Kleiner, T., and Humbert, A.: Results of "Comparison of ice dynamics using full-Stokes and Blatter-Pattyn approximation: application to the Northeast Greenland Ice Stream" (Version v1) [Data set], Zenodo, https://doi.org/10.5281/zenodo.6406103, 2022.

Saad, Y.: Iterative Methods for Sparse Linear System, Philadelphia, PA, USA: Society for Industrial and Applied Mathematics, 2003.

Seddik, H., Greve, R., Zwinger, T., Gillet-Chaulet, F., and Gagliardini, O.: Simulations of the Greenland ice sheet 100 years into the future with the full Stokes model Elmer/Ice, Journal of Glaciology, 58, 427–440, https://doi.org/10.3189/2012JoG11J177, 2012.

Seddik, H., Greve, R., Zwinger, T., and Sugiyama, S.: Regional modeling of the Shirase drainage basin, East Antarctica: full Stokes vs. shallow ice dynamics, The Cryosphere, 11, 2213–2229, https://doi.org/10.5194/tc-11-2213-2017, 2017.

Seroussi, H., Nowicki, S., Simon, E., Abe-Ouchi, A., Albrecht, T., Brondex, J., Cornford, S., Dumas, C., Gillet-Chaulet, F., Goelzer, H., Golledge, N. R., Gregory, J. M., Greve, R., Hoffman, M. J., Humbert, A., Huybrechts, P., Kleiner, T., Larour, E., Leguy, G., Lipscomb, W. H., Lowry, D., Mengel, M., Morlighem, M., Pattyn, F., Payne, A. J., Pollard, D., Price, S. F., Quiquet, A., Reerink, T. J., Reese, R., Rodehacke, C. B., Schlegel, N.-J., Shepherd, A., Sun, S., Sutter, J., Van Breedam, J., van de Wal, R. S. W., Winkelmann, R., and Zhang, T.: initMIP-Antarctica: an ice sheet model initialization experiment of ISMIP6, The Cryosphere, 13, 1441–1471, https://doi.org/10.5194/tc-
665      13-1441-2019, 2019.

Seroussi, H., Nowicki, S., Payne, A. J., Goelzer, H., Lipscomb, W. H., Abe-Ouchi, A., Agosta, C., Albrecht, T., Asay-Davis, X., Barthel, A., Calov, R., Cullather, R., Dumas, C., Galton-Fenzi, B. K., Gladstone, R., Golledge, N. R., Gregory, J. M., Greve, R., Hattermann, T.,

Hoffman, M. J., Humbert, A., Huybrechts, P., Jourdain, N. C., Kleiner, T., Larour, E., Leguy, G. R., Lowry, D. P., Little, C. M., Morlighem, M., Pattyn, F., Pelle, T., Price, S. F., Quiquet, A., Reese, R., Schlegel, N.-J., Shepherd, A., Simon, E., Smith, R. S., Straneo, F., Sun, S., Trusel, L. D., Van Breedam, J., van de Wal, R. S. W., Winkelmann, R., Zhao, C., Zhang, T., and Zwinger, T.: ISMIP6 Antarctica: a multi-model ensemble of the Antarctic ice sheet evolution over the 21st century, The Cryosphere, 14, 3033–3070, https://doi.org/10.5194/tc-14-3033-2020, 2020.

Steinemann, S.: Results of Preliminary Experiments on the Plasticity of Ice Crystals, Journal of Glaciology, 2, 404–416, https://doi.org/10.3189/002214354793702533, 1954.

Urquiza, J. M., Garon, A., and Farinas, M.-I.: Weak imposition of the slip boundary condition on curved boundaries for Stokes flow, Journal of Computational Physics, 256, 748–767, https://doi.org/10.1016/j.jcp.2013.08.045, 2014.

van der Veen, C. and Whillans, I.: Force Budget: I. Theory and Numerical Methods, Journal of Glaciology, 35, 53–60, https://doi.org/10.3189/002214389793701581, 1989.

Vanka, S.: Block-implicit multigrid calculation of two-dimensional recirculating flows, Computer Methods in Applied Mechanics and Engineering, 59, 29–48, https://doi.org/10.1016/0045-7825(86)90022-8, 1986.

Verfürth, R.: Finite element approximation on incompressible Navier-Stokes equations with slip boundary condition, Numerische Mathematik, 50, 997–721, https://doi.org/10.1007/BF01398380, 1986.

Widlund, O. and Toselli, A.: Domain decomposition methods - algorithms and theory, vol. 34 of *Springer Series in Computational Mathematics*, Springer, 2005.

Zhang, T., Ju, L., Leng, W., Price, S., and Gunzburger, M.: Thermomechanically coupled modelling for land-terminating glaciers: a comparison of two-dimensional, first-order and three-dimensional, full-Stokes approaches, Journal of Glaciology, 61, 702–712, https://doi.org/10.3189/2015JoG14J220, 2015.