# Peer review of "Comparison of ice dynamics using full-Stokes and Blatter-Pattyn approximation: application to the Northeast Greenland Ice Stream"

_The Cryosphere, 2021_

## Author Response (AR1)

Dear Editor and Reviewers,

The authors wish to thank both reviewers for careful reading of the manuscript, helpful remarks and constructive criticism. We highly appreciate the suggestions and follow most of the comments to draw a more reliable conclusion. Since the criticisms by the reviewers touched (partly) the same issues we decided to add more experiments to the manuscript. The criticisms were mainly related to (1) not covering a grounding line region, (2) type of friction law, (3) numerical robustness, (4) the study of englacial layering is too limited and (5) the time-independence. According to these points, we decided to strengthen the paper by focusing more on points (1) to (3). In doing so, the main changes are as follows:

1) We included a second domain (cyan line in the Figure R1) that covers the grounding line region of 79NG and calving front of ZI.

[Figure]

*Figure R1: Overview of the NEGIS region and modelling domain. (a) Observed surface velocities (Joughin et al., 2016, 2018). (b) Bed topography (Morlighem et al., 2017). The red and cyan line in (a) and (b) delineates the 'ice-stream' and 'outlet' modelling domain. White lines in (a) indicate flux gate locations at 79NG and ZI.*

2) We agree that the type of friction law (linear (m=1) or power law (e.g. m=3)) might be critical for such an analysis. Therefore, we conducted some additional experiments with a power law (m=3) and added them to the manuscript.

3) To test the numerical robustness of the simulations we employ different discretization schemes, namely P1+P1 stabilized with GLS (the one used in the first version of the manuscript) and P2+P1 (TaylorHood); the latter is independent of stabilization parameters. We also test two different implementations of the friction law (called strong and weak). The latter was not a request by the reviewers, but we think it is part of such a 'robustness' test.

Based on these extensions we found that BP-like area-averaged deviations are larger (up to ~17%) to FS compared to the moderate differences (~1.5%) we found in the 'ice stream' domain for E=1 (Figure R2). For m=1 we found a smaller response as for m=3. In general, the different discretization schemes and friction implementations point into the same

directions but with different amplitudes. However, we hope to convince the reviewers by these experiment extensions that our study has a broader view.

[Figure]

**Figure R2:** *Relative surface velocity differences of spatially averaged surface velocities from FS and BP-like stress regimes. (a) Linear friction law (m=1). (b) Non-linear friction law (m=3).*

Regarding points (4) and (5):

4) We fully agree that the englacial advection analysis is simple (stationary) and very limited. It shouldn't provide accurate and realistic layer elevations (e.g to be compared with radar isochrones) but it was intended to give a feeling about the expected error on englacial layers by employing different stress regimes. Due to the fact that internal layer information is important for the interpretation of deep ice cores or for calibration of ice sheets model, we thought such a rough estimation would be useful although it is simple. On the other hand, this analysis demonstrates how differences between FS and BP-like could evolve over time. It is a rough analysis and needs to be confirmed with further works. Moreover, to our knowledge, it is the first time that particle pathways of a BP and FS stress regime were compared. However, we decided to drop this part due to the reviewer's criticism and due to the expected manuscript length, particularly as we added new material to the manuscript.

5) We agree that the time-independence of our simulations is a major drawback. However, the focus was on studying dynamical differences between FS and BP under identical numerical and physical conditions (similar as in the ISMIP-HOM experiments or as in Morlighem et al (2010)). We are analyzing this in a realistic setup with a high-resolution (up to 100m horizontal resolution). To our knowledge, such an analysis was not performed before. We think that our analysis demonstrates that a really high resolution is needed to bring full-Stokes effects into account. Although we have resolutions up to 100m our simulations indicate that FS is not 'converged' and differences to BP-like are still increasing.

The first version of the manuscript concluded that FS is not urgently needed for projections of the Greenland ice sheet compared to other uncertainties in ice sheet modelling. The conclusion was somewhat misleading. It should rather be concluded that "FS is not urgently

needed for fast ice-stream flow like NEGIS". However, based on the new simulations and findings we have to adjust our conclusions:

1.  FS is essential near the grounding line of 79NG but of less importance for ice stream flow.
2.  We found the basal drag as the major source of the differences. Although we found an equivalent response as Morlighem et al. (2010) we are not able to relate it to the bridging stresses. Morlighem et al. (2010) concluded that FS is needed on retrograde beds. In our setup the geometry is different as the bed rises inland at 79NG and ZI. See detailed answer to comment 'l. 367' by Reviewer #1.

Technicalities: below we answer each point raised by the reviewers (reviewer text = black text) and mark our answer in blue color. 'Done' denotes that this point would be solved in the revised version of the manuscript. This could be that it will be either done directly, or that due to other changes the point does not arise any more.

---------------------------------- **Reviewer #1** ----------------------------------
**General comments**

This study addresses a longstanding question in ice sheet modeling: Are there regions where full-Stokes (FS) models are much more accurate than the Blatter-Pattyn (BP) and other higher-order (HO) approximations, and therefore FS models are needed for accurate sea-level projections? By implementing FS and BP-like solvers using the COMSOL finite element package, the authors have developed a useful tool for exploring differences between FS and BP, without complications due to differing numerics. They apply these solvers to the central part of the Northeast Greenland Ice Stream (NEGIS) at a wide range of resolutions (0.1 km to 12.8 km) and find modest differences between FS and BP. As expected, the differences are greatest at fine resolution and in regions with a high slip ratio, high aspect ratio, and/or rough topography. Differences are small at resolutions of 1 km or coarser, as typically used in ice sheet models. Since these differences are small compared to other uncertainties in ice sheet modeling, the authors conclude that FS models are not urgently needed for sea-level projections.

The question is an important one and is not settled. Morlighem et al. (2010) argued that when inverting for basal drag coefficients in ISSM, bridging effects near the grounding line of Pine Island Glacier (PIG) make it essential to use FS models. Other authors, including Nowicki and Wingham (2008) and Durand et al. (2009), have made similar arguments. Today, many ice sheet models with HO approximations are being used for Greenland and Antarctic projections on century time scales, while FS models are used sparingly because of their complexity and expense. As far as I know, the projections based on FS models, including Elmer/Ice, are not dramatically different from projections based on HO models. I am inclined to agree with Ruckamp et al. that FS–BP differences are secondary compared to other uncertainties, but it is still important to explore these differences systematically.

The authors take a step in this direction with their detailed NEGIS analysis, but this step does not go far in settling the question. They focus on part of a single ice stream without addressing other regions, such as PIG, where stress-balance terms neglected by BP could be important. As the authors acknowledge, their analysis does not include prognostic simulations or thermomechanical coupling, and they have tested only one basal sliding law. So the analysis does not strongly support the conclusions.

I would not expect these complex issues to be resolved in one paper, but I would like to see a broader analysis. For example, the authors could simulate an entire ice sheet in a transient run of a few years at the highest affordable resolution. Or they might look at regions where others have argued that FS models are needed. Perhaps this is impractical

for the COMSOL solver and would require a different model such as ISSM. But I would like a sense that having developed the COMSOL tool, the authors have pushed it as far as they can. I encourage them to consider other applications that would strengthen their main argument.

We would like to thank the reviewer for raising these points, it really helps to draw a better conclusion. Within the new version of the manuscript, we have now included a second region that includes the grounding line of 79NG and the calving front of ZI. We also varied the type of friction law (linear and power-law). As recommended by the other reviewer we also test the numerical robustness in different ways. We hope to convince the reviewer with these applications that we can draw a more reliable and broader conclusion compared to the first version of the manuscript.

Also, the paper needs significant editing for correct English.

**Specific comments**

Abstract, l. 12: "severe impacts on internal layers of ice sheets". The discussion of englacial advection is limited and does not justify this strong and rather vague statement in the Abstract.
The section about englacial advection was intended to illustrate that small initial FS-BP-like differences could be built up over time. It shouldn't be interpreted as a 'real' layer model. However, both reviewers found this analysis too limited. With expanding this section, we think the paper will be too long, especially as we add more material (new 'outlet' domain, numerical tests, power-law friction). Therefore, we decided to drop this part from the manuscript.

l. 15: "no simplification". This is a bit strong; all model equation sets have some simplifications (e.g., isotropic, temperature-dependent flow factors in FS models).
We dropped that part of the sentence.

l. 26: Please cite the papers by Blatter and Pattyn here rather than below at l. 62.
Done.
Also, I think "severe" should be "several". (Maybe this was also true in l. 12?)
Changed as suggested. Line 12 is dropped in the new version of the manuscript.

l. 30: "only one contribution". I think this is the Elmer/Ice model? Please include the model name and a citation.
Correct, this is Elmer/ice. We added in the corresponding-citation "*model UTAS_ElmerIce with a variable resolution between 4 and 40km*".

l. 39: "with the analytical solution". I don't think the ISMIP-HOM experiments have analytical solutions. The FS results serve as a benchmark to which HO models are compared.
This is not correct. ISMIP-HOM provides an analytical solution for one experiment (ISMIP-HOM Exp. F.; compare the last sentence of the abstract of Pattyn et al (2008) and the following text). However, we changed the sentence to: *"...and with one available analytical solution."*

l. 40: "is prohibited". This is not the right word. Maybe "is not possible"?
Changed as suggested.

l. 43: "huge additional computational amount". Is it possible to give an order-of-magnitude estimate of the additional cost?
With our implementation there is no computational benefit of the BP-like model compared to FS. The classical BP scheme shows lower computational costs than BP. The inversion of

basal friction for the Greenland Ice Sheet in Larour et al. (2012, Table 3) shows that the classical BP is a factor of about 10 faster. Both setups FS and BP were run under identical conditions (mesh, solver, cluster settings). We added: *"... given the huge additional computational cost by FS models (about 10 times slower than BP (Larour et al., 2012, Table 3 therein))."*

ll. 45–55: This paragraph mentions some studies that used FS, but it would be helpful to the reader to give more details, for example the argument of Morlighem et al. (2010) that FS is needed near Antarctic grounding lines.
We added to the text: "*Morlighem et al., (2010) concluded that treating ice flow of fast-flowing glaciers with a steeply rising bed near the grounding line with FS is essential.*"

Of the three difficulties listed here for FS–BP comparisons, only (i) is addressed by the present study. I wish the authors had pressed their analysis further; see the general comments.
First of all, we would like to clarify that the point (ii) is also addressed in our study. We use the same type of basal flow condition for FS and BP-like to make a fair comparison. In Favier et al. (2014), the model UA uses a power law (m=3) while Elmer/Ice and BISICLES relying on a linear law (m=1). This makes a comparison difficult.
However, in the new manuscript we present two different types of friction laws: linear (m=1) and a power law (m=3).

l. 52: "which resolves, e.g. bed differently". The meaning here is not clear.
We have rewritten the sentence to: *" … how much of the difference is due to numerical treatments as, e.g. different horizontal grids resolve the bed differently …".*

l. 60: "certainly because FS and higher-order models are too expensive for these long-time integrations". Some HO models, especially depth-integrated models (e.g., Goldberg 2011), are, in fact, practical for long time integrations.
Well, for studying the interior of an ice sheet (vertical advection, layering etc) we need a 3D model. However, this part of the paper is dropped.

l. 67: "Therefore…" This is not a strong argument for focusing on a subset of NEGIS. Does this region have characteristics that could make it especially challenging for BP models as opposed to FS? The text mentions bed topography, and later slip and aspect ratios, but it is unclear that central NEGIS is an appropriate analog for, say, the PIG grounding line or for glaciers with rougher topography.
Our intention was not to run an analog to PIG. Our main intention was to run a clear comparison between FS and BP in a region where we expect that BP might fail. Therefore, we selected the NEGIS ice stream where we have a variable topography with smooth and rough bed gradients (see Fig. 3). Beside this challenging characteristic for BP, we expect a high-slip ratio at the ice stream. Based on the works by Hindmarsh (2004) and Gudmundssopn (2003) we know that BP has the potential to fail in these areas. Generally challenging (independent of area) is the increased aspect ratio with higher resolution so that higher order stress terms might be no longer negligible. However, due to the changes we made to the manuscript (see points 1-3 in the preamble), we have rewritten this part: "*We select two subsets of the Northeast Greenland Ice Stream (NEGIS) as an investigation area. Both subsets cover slow and fast flowing ice as well as a smooth and highly variable bed topography. The first domain focuses on the ice stream while the second domain covers the NEGIS outlets 79°N Glacier (79NG) and Zacharias Isbrae (ZI).*"

l. 74: "ratio between basal sliding and gravitationally driven flow". I think this should be "basal sliding and internal deformation".
Done.

l. 100: "usually includes the effective pressure". I suggest "often includes". Weertman-type power laws, for example, do not include effective pressure.
Done.

ll. 130–140. I like this approach to coding FS and BP in a way that isolates the discarded stress terms and minimizes differences in model numerics.
Thanks! See also answer to line 367. It shows that the ice flow model is very flexible to study various ice flow components (e.g., we realized the LTSML stress regime (this is roughly BP plus bridging stresses) and an FS regime without bridging stresses).

l. 164: "limiting case". I think what is meant here is that each solver has a range of resolutions where it is applicable, and 0.4 km lies in the overlap region. Please clarify; it would help to state the range of applicability for each solver.
We would avoid making a general statement about the range of applicability of the solvers regarding the problem sizes. The wording "limiting case" is a bit misleading here. Generally, the solvers are applicable to each problem size. But the iterative solver shows a poor performance when applied to the small problems (but still converge) while MUMPS run out of memory for the larger problems. The latter might not be a problem on clusters which are equipped with more memory per physical node. So, we aimed to apply a direct or iterative solver where they show the best performance. We changed the paragraph accordingly:
*"Regarding the desired consistency between the simulation results, we must clarify the different usage of the linear solvers:the coarser models show a poor performance with the ASM solver. At the same time, the higher resolutions run out of memory with the MUMPS solver. Dependent on modelling domain and employed discretization MUMPS run out of memory around 5 million DOF's. In that limiting case MUMPS and ASM reveals differences between well below 10^−7 m a^−1 in the surface velocity; differences between MUMPS and Vanka below10^−3 m a^−1. Consequently, we assume that all applied linear solvers provide comparable results."*

l. 188: "On purpose…". Why was a region chosen far upstream of the grounding line, given that grounding lines are important for sea-level projections and might be regions where FS–BP differences are large?
We intended to look at an ice stream rather than at the grounding line. With the employed high-resolution of 100m, such an analysis was not performed before to our knowledge. In the new manuscript, we have introduced another (second) region that focuses on the grounding line (79NG) and calving front (ZI).

l. 193: Why a Budd-like friction law? Most ice sheet models are now using a power law, a Coulomb law, or some hybrid combination. See, e.g., Sect. 2.1 of Asay-Davis et al. (2016). In general, the analysis would be more compelling if it included more than one basal friction law.
The Budd-like of friction law is often used in ice sheet modelling (Morlighem et al., 2010; Price et al., 2011; Gillet-Chaulet et al., 2012; Seroussi et al., 2013, Choi et al., 2021), it implies that the basal drag can increase unbounded. It was shown that inducing an upper ratio of τb/N (Iken's bound) is more justified (Iken, 1981; Schoof, 2005; Gagliardini et al., 2007; Leguy et al., 2014; Joughin et al., 2019). We choose the linear friction law (m=1) as it is commonly used in ISMs making use of inverse methods to constrain the basal friction (Morlighem et al., 2010; Larour et al., 2012; Seroussi et al., 2013; Perego et al., 2014; Gladstone et al., 2014). In the new manuscript version, we have now included an analysis with a power law (m=3).

l. 209: What is meant by a symmetry BC? Which field or fields are symmetric across the boundary?
That means no flow across the ice boundary is allowed and shear stresses are vanishing. We have slightly rewritten this part to:

*"Solving a subset of an ice stream poses unknown boundary conditions in the interior of the ice sheet. A very simple approach would be prescribing the measured surface velocities as a depth-averaged velocity profile. However, we choose boundary conditions that are free to adjust during the solution process. Laterally boundary conditions are chosen to have a symmetry boundary condition at the inflow boundary, which implies v\*n=0 and vanishing shear stresses. A free slip condition is chosen at lateral along-flow boundaries and a normal stress condition at the outflow boundaries. Dependent on the setup, the outflow boundaries are land- or marine terminating fronts of the glacier or located within the ice. For the former, we prescribe t·n= min(rho_i \* g (z_s−z),0), where z_s is the surface elevation and z the vertical coordinate. For the latter, we impose t\*n=rho_i\*g (z_s−z) +δ. Here, δ is an additional stress that is slightly tuned to match the observed velocities at those boundaries; it is in the order of 0.1 MPa."*

l. 212: The wide range of resolutions is a strong point of this study.
Thanks.

l. 220: "very extreme". I'm not sure why the authors chose some unrealistic values. I would suggest three values: E = 1, plus values that are on the low side and the high side but still physically plausible (say, 0.5 and 3). Below, when there are FS–BS differences with E = 0.1 or E = 6, it is hard to know whether to take those differences seriously.
We agree that one has to take the results from the extreme E values with caution. We consider these extreme values in order to understand the behaviour from stiff to hard ice. When choosing values that are much closer to E=1, the responses are likely too similar in our setup; this would not help to understand the general model behaviour. We think for such a sensitivity study these extreme values are justified. Therefore, we intend to keep our suggested range.
However, values like E=~5 and E=~0.5 (rather for SSA i.e. ice shelf flow) are used in the community (e.g., de Boer et al.; 2015, Ma et al., 2017). In the updated version of the manuscript, we would refer to these values as extreme but not as unrealistic.

l. 256: "up to 43 m/a compared to FS". To give readers a sense of the percentage error, please state the FS value.
Done. We also did this for the following values.

l. 260: "for very soft ice…". Since E = 6 may be unrealistic, the significance of these differences is unclear. Similarly for the stiff-ice case, l. 273. See the comment above.
See answer to 'line 220'.

l. 273: "Maximum differences…". Is this in relative rather than absolute terms?
This is in relative terms. We changed it to "Relative maximum differences …".

ll. 315ff: I don't think Sect. 5.5 adds much to the paper. I would guess that the FS–BP differences are small compared to the vertical diffusion that would be associated with remapping variables onto a modest number of layers, but this could only be shown in a prognostic run. I would drop this section if it isn't possible to say more.
See answer to "Abstract, l. 12:".

l. 330: "whether FS or BP-like is located below or above each other". Please clarify what this means.
Done. See answer to second reviewer.

l. 351: "particularly in extreme cases…" See comments above about unrealistic E values.
See answer to 'line 220'.

l. 359: "It might be favorable…" I agree with the statement, but it is not strongly supported by the single example. See general comments.

We have now included a second region that covers the grounding line of 79NG and the calving front of ZI. See answer above.

l. 361: "our simulations are not prognostic". As stated above, this is an important limitation. Analyzing a prognostic problem, if possible, would strengthen the paper.

See comment above.

l. 367: The authors cite Morlighem et al. (2010), but that paper draws a different conclusion (that FS models are essential). Does the NEGIS analysis cast any doubt on the Morlighem conclusions?

The reviewer is right with raising this point. We somehow observe a similar behaviour between FS and BP as in Morlighem et al. (2010). They did an inversion to match the observed velocities. They found that the FS model needs to reduce the basal drag compared to BP. They attribute this behavior to the developing bridging stresses in FS due to the rising bed close to the grounding line.

In our simulations we have a different setup with no constraint on the surface velocity. The FS model develops a higher basal drag than BP-like and in turn lower velocities. This is an equivalent behaviour as in Morlighem et al. (2010). However, in our simulations the response is very small for E=1.

With new simulations we observe a similar behaviour but velocity differences between FS and BP-like are much stronger than in the 'ice-stream'-setup; especially at the grounding line of 79NG (up 400m/a difference (~44%)). In Morlighem et al. (2010), the rising bedrock towards the grounding line leads to higher sigma_zz in FS compared BP (about 2%), which in turn causes the reduction in basal drag. However, in our simulations we don't find a clear connection to bridging stresses. Figure R3a shows the relative differences between FS-sigma_zz and BP-like-sigma_zz. Although FS-sigma_zz shows differences to BP in a similar range (+/- 2%) as in Morlighem et al. (2010) there is no clear trend. However, in Morlighem et al. (2010) the bed rises towards the grounding line, while at 79NG the bed slopes down towards the grounding line. In our case the differences stem from the basal drag (Figure R3b) that is in the majority higher in FS compared to BP-like. Those regions overlap where the assumption that horizontal gradients of the vertical velocity are small compared to the vertical gradient of the horizontal velocity is invalid (Figure R3c). Those terms are of similar order. Therefore, dropping d_x v_z and d_y v_z led to a lower basal drag in BP-like. Consequently, we assume that sliding is overestimated in BP-like. Based on the new simulations we come to the same conclusion as Morlighem et al. (2010) that FS is essential at the grounding line.

[Figure]

*Figure R3:* *Comparison of simulations results for a resolution of 150m, m=3 and P1P1GLS-weak. (a) Relative difference of sigma_zz at the ice base between FS and BP-like. (b) Absolute difference of tau_b between FS and BP-like. (c) Ratio of vertical gradient of the horizontal velocity to horizontal gradients of the vertical velocity in absolute terms from the FS solution.*

In order to highlight that the bridging stresses play a minor role in our setup, we have run two other stress regime versions (executed with P1P1GLS-weak). The first one keeps the bridging stresses in the third equation of the momentum balance (this is similar to LTSML according to Hindmarsh (2004)); the second one is a FS model without bridging stresses. The area averaged results reveal that LTSML-like is close to BP-like while FS wo bridging stresses is rather similar to FS (Figure R4). These simulations also demonstrate the high flexibility of our ice flow model to test various ice flow components.

[Figure]

*Figure R4:* *Relative surface velocity differences of spatially averaged surface velocities from FS and BP-like stress regimes for m=3. Compared to Figure R2b, we also show two simulations from an*

*LTSML-like and 'FS without (wo) bridging stresses' regimes simulated with P1P1GLS-weak. Relative differences are calculated between FS and LTSML or FS wo bridging stresses.*

l. 375: "there are indications that small initial differences become much larger over long time integrations." It is good to acknowledge a study's limitations, but this is another example of the analysis being too limited to draw broad conclusions.
We hope to convince the reviewer with the new simulations (new domain, power-law friction, numerical tests) that our study is not too limited.

l. 382: "The model disagreements still tend to diverge below…". This wording is unclear. Maybe "The models still do not agree at…"
Done. The sentence is rewritten.

l. 387: "a view on particle pathways…" Again, I think a diagnostic run is not sufficient to draw conclusions on englacial layering.
See answer to "Abstract, l. 12:"

l. 391: "FS will start to matter…". This has been shown only for regions that are like NEGIS in relevant ways, where there are no differences introduced by long-term advection, thermodynamic evolution, or grounding lines.
Done. Sentence is rewritten.

l. 395: "the use of FS seems not an urgent issue." See the general comments.
See answer to "Abstract, l. 12:"

Tab. 1: This is a short list of constants. Are there any others?
We dropped the table with the list of constants. The values are now stated in the text.

Fig. 4: The inset panels in the upper left of each panel are not described in the caption and are hard to read. Possibly expand to full panels with a separate caption.
We dropped the upper left panels. The higher spread of E=6 compared to E=1 is already visible from the map and also shown in Fig. 5.

Fig. 5: The axis labels are hard to read.
We have increased the labels.

The figures labeled as A2 to A10 are not related to Appendix A. These should be included instead as supplementary material.
Done. We included a supplement.

**Technical corrections**

The paper contains many grammatical errors and uses of non-standard English. This is a partial list.

Title: Usually I have seen "Northeast" rather than "North East" for NEGIS
You are right. We changed 'North East' to 'Northeast'.

l. 2: "its owing" is not idiomatic
The sentence is rewritten to *"However, their applicability is often limited due to the high computational demand numerical challenges."*

l. 3: "consequences caused by" is redundant
Done

l. 7: "increases" -> "increase"
Done

l. 15: "is by using" -> "is given by"
Done:

l. 26: "computational" -> "computationally"
Done

l. 32: "fidelity to accurately simulate" is awkward. Maybe "ability to accurately simulate"
Done.

l. 43: "computational amount consumed by" -> "computational cost"
Done

l. 55: "processes and interactions that **make** it difficult"
Done.

l. 68: "higher variable" -> "highly variable"
Done.

l. 74: "the enhancement factor" -> "an enhancement factor"
Done.

l. 83: "isometric" -> "isotropic"
Done

l. 99: "outwards of" -> "out of"
Done.

l. 109: "explained below" -> "as explained below"
Done.

l. 110: "are forming" -> "form"
Done

l. 122: "simplifications … reduces" -> "simplifications … reduce"
Done.

l. 126: "are following" -> "follow"
Done.

l. 147: "computational amount" -> "computational cost"
Done.

l. 188: "upstream **from** the grounding line", "downstream **from** the ice divide"

l. 193: "friction type" -> "friction law"?
Done.

l. 232: "towards" -> "and", "consumed computational resources" -> "computational costs"
Done.

l. 236: "compared exemplary" -> "compared"
Done.

l. 244: "excepted" -> "expected"
Done.

l. 264: "unveils" -> "shows" or "reveals"
Done.

l. 267: "show increasing trends" -> "increase"
Done.

l. 293: "leveled out" -> "compensated"
Done.

l. 309: "less" -> "fewer"
Done.

l. 378: "alleviate" -> "allow" or "enable"
Done

l. 379: "issues" -> "differences"
Done.

l. 393: "uncertainties" -> "uncertainties in"
Done.

Tab. 2 caption:  "Number for" -> "Number of",  "exemplary listed" -> "listed"
Done.

**References**

Davis, X. S., et al., Experimental design for three interrelated marine ice sheet and ocean model intercomparison projects: MISMIP v. 3 (MISMIP+), ISOMIP v. 2 (ISOMIP+) and MISOMIP v. 1 (MISOMIP1) (2016), Geosci. Model Dev., 9, 2471–2497, doi:10.5194/gmd-9-2471-2016.

Durand, G., O. Gagliardini, B. de Fleurian, T. Zwinger, and E. Le Meur (2009), Marine ice sheet dynamics: Hysteresis and neutral equilibrium, J. Geophys. Res., 114, F03009, doi:10.1029/2008JF001170.

Goldberg, D., A variationally derived, depth-integrated approximation to a higher-order glaciological flow model  (2011), J. Glaciol, 57, 157–170.

Morlighem, M., E. Rignot, H. Seroussi, E. Larour, H. Ben Dhia, and D. Aubry (2010), Spatial patterns of basal drag inferred using control methods from a fullâStokes and simpler models for Pine Island Glacier, West Antarctica, Geophys. Res. Lett., 37, L14502, doi:10.1029/2010GL043853.

Nowicki, S. M. J., and D. J. Wingham (2008), Conditions for a steady ice sheet–ice shelf junction, Earth Plan. Res. Lett., 265, 246-255.

----------------------------------- **Reviewer #2** -----------------------------------
Review Josefine Ahlkrona

**General comment**

The paper addresses the question whether the computationally expensive full Stokes equations are really necessary, or if the cheaper Blatter-Pattyn model is sufficient. They do so by simulating a part of the NEGIS ice stream with both models using the same code (COMSOL) and as similar numerical discretization as possible. The results are then compared using a series of different measures. The experiments show that unless the ice is very stiff, the difference in velocity field amounts to a few percent, but that there is potentially a larger difference in internal ice deformation, which may have implications for paleo reconstructions.

The topic is a very important one and this type of paper is needed. I also appreciate that the authors measure the error in several different ways. However, the study is limited, and I am afraid that the reader might draw the overhasty conclusion that full Stokes is not needed based on these results, while there might be other situations where full Stokes is important. The main limitation as I see it is:

- The lack of a grounding line experiment: I understand that it might be difficult to implement grounding line migration in a commercial software, but I think at the very least the authors can look at a time-independent grounding line problem, comparing velocity fields and buoyancy balance.
  In the new version of the manuscript, we have included a second domain called 'outlet'. It includes the grounding line of 79NG and the calving front of ZI. Very briefly, within this domain we see a higher velocity difference between FS and BP-like, and consequently, higher ice discharge for BP-like. We also looked at the buoyancy balance (we evaluated the contact condition -sig_nn|b>p_w (e.g., Durand et al. 2009). In this stationary problem, BP-like reveals a grounding line that lies further upstream (~0.5 km) than the one calculated from FS. But both are located within the hinge zone of 79NG.

Also, I think the authors should consider the following issues:

- The lack of time dependence: The study does not include time dependent simulations. The impact of the model differences on surface evolution can be measured without actually running a surface evolution, I think it would suffice to look at how the velocity field would change the surface after one time step. However, I wonder if it could be a problem that the initial surface has not been relaxed. Are we looking at an artificial initial shock transient? Is that relevant?
  We agree that the lack of time dependence is a drawback of our study. However, for this study it was not the primary focus. We intended to evaluate the ice dynamics in a stationary setup by keeping all other settings identical (similar as in ISMIP-HOM or (somehow) as in Morlighem et al 2010). In addition, a comparison with such a high-resolution of 100m in a realistic setup was not performed before. It is also not very clear how to differentiate between an 'initial transient shock' and the effect of model physics. What would be the characteristics of such a shock? We followed the reviewer's recommendation and computed the free surface evolution after one time step via the kinematic boundary condition assuming no melting or accumulation (also called emergence velocity, $\mathbf{v}*\mathbf{n}$). In Figure R5a we show the emergence velocity for FS and the difference between FS and BP-like. The absolute field shows values of about +/- 50m/a (a bit higher at some isolated grid nodes), which are high but not very extreme (assuming an initial time step of 0.05 years that corresponds roughly to +/- 2.5m height changes at the first time step). Assuming ongoing elevation changes (dz_s/dt) in that area of about 1 to 2 m/a (derived from Laser Altimetry, pers. comm. Veit Helm, AWI, and older data published in Helm et al., 2014) we have an estimated maximum ratio (emergence velocity/(dz_s/dt)) of about 25. We don't find good arguments whether or not those emergency velocities indicate an initial shock. We would of course like to incorporate quantitative/objective measures related to this topic, but we don't find any in the existing literature. However, our subjective view is

that the signal from the emergence velocity is not dominated by a so-called initial shock as it is in a reasonable range (for FS about +/- 50m/a).

I would like to see at least a discussion justifying the lack of relaxation.

Unfortunately we don't see a way to keep the simulations consistent by performing a relaxation run for BP and FS. As consistency was the primary target of the model set-up to study the response of the model (with different physics) to different initial and boundary conditions, we run the model without initial relaxation. One could argue for a relaxation run based on e.g., the BP-like scheme and run successive simulation in BP-like and FS, but then the FS would experience a similar type of 'shock' because the geometry would be consistent only to the BP-like physics (and in the same way also for a FS relaxation run). Performing relaxation simulations for BP-like and FS individually would result in different geometries and also different initial conditions (e.g., velocity) to start with. This would be an undesired effect for our consistency approach which would make a clear comparison difficult. Here, we followed the strategy to have physical parameters and the geometry input equal in the FS and BP-like to allow a better comparison. We will make this much clearer in the updated version of the manuscript.

[Figure]

**Figure R5:** *Evaluation of the kinematic boundary condition (emergence velocity **v***n*) at the surface for l=150m, m=3, and P1P1GLS-weak. Surface mass balance is assumed to be zero. (a) FS emergence velocity. (b) Emergence velocity differences between FS and BP-like (**v***n|FS - **v***n|BP-like).*

- The boundary conditions are retrieved by inverting with BP (and a different code). What does it mean for FS that the boundary conditions are consistent with another model? It would be interesting to see another set of experiments, where the slip coefficients are retrieved using FS (e.g. with Elmer) and are then used for both FS and BP. Also, would inverting at a higher resolution make a difference? Would there be high frequency effects that FS would pick up?

This is an excellent point raised! The original idea was to study the differences between FS and BP-like under identical basal conditions. To test whether the results are sensitive to different basal friction fields. Therefore, as the reviewer suggested, we did now run an ISSM inversion with BP with a resolution of 200m and an ISSM inversion with FS with a resolution of 800m. We fed the inferred friction to COMSOL and ran the simulations up to a resolution of l=400m with the P1P1GLS discretization scheme and the strong implementation of the friction law. We also choose the new domain 'outlet'. The relative error of the area-averaged surface velocities between FS

and BP-like show almost no differences (Figure R6). In the discussion about numerical robustness, we will now refer to it.

[Figure]

*Figure R6: Relative surface velocity differences of spatially averaged surface velocities from FS and BP-like stress regimes for m=3 and P1P1GLS-strong. The coloured lines indicate the inferred friction coefficient with ISSM used as input for COMSOL.*

- Some tests regarding the numerics are missing. Since quite some effort is taken to treat the models with similar numerics, I would like to see some tests or discussion convincing the reader that the discretization does indeed not impact the result, since the interest is in quite small velocity differences. In particular, the inf-sup stabilization parameter may not be the same for FS and BP (it is not clear if the same stabilization is used for the BP system, but I assume so), and the problem could, if you are unlucky, be sensitive to this. Either change to Taylor-Hood elements or check that varying the stabilization parameter for the inf-sup stabilization does not alter results. Also, the element aspect ratio varies in the experiments, as the number of vertical layers are constant. Will the numerical errors of FS and BP behave the same when element aspect ratio changes? Perhaps this is not relevant but if so, a comment on why should be included.

  Again, this is an excellent point raised! We have now included a couple of simulations to test the numerical robustness. We used different discretization schemes (P1P1+GLS and P2P1 as well as weak and strong imposition of the friction law (e.g., Figure R2, R4 and R8). Also, we did a test on the element aspect ratio by varying the number of vertical layers. For this test we ran the simulations up to a resolution of l=400m with the P1P1GLS discretization scheme and the strong implementation of the friction law. We also choose the new domain 'outlet'. It seems to be a very minor issue (Figure R7). In the discussion about numerical robustness, we will briefly refer to it.

[Figure]

*Figure R7: Relative surface velocity differences of spatially averaged surface velocities from FS and BP-like stress regimes for m=3 and P1P1GLS-strong. The coloured lines indicate the number of vertical layers.*

- The idea of studying internal layers (section 5.5) is nice, but this part of the study is too limited.
  The section about englacial advection was intended to illustrate that small initial FS-BP-like differences could be built up over time. It shouldn't be not interpreted as a 'real' layer model. However, both reviewers found this analysis too limited. With expanding this section, we think the paper will be too long, especially as we add more material (new 'outlet' domain, numerical tests, power-law friction). Therefore, we decided to drop this part from the manuscript.

**Specific comments**

Line 30 - worth to mention that that FS simulation was with coarse resolution
Done. We added in the citation: "*model UTAS_ElmerIce with a variable resolution between 4 and 40\,km*".

Line 76 - Consider changing the title "field equations" to e.g. "the full stokes equations"
Done. We changed the title as suggested.

Section 2.3 Explain to the reader in what situation all of the neglected stress components are important. Which are important for shearing margins, which are important at grounding line, etc.
You are right, this may help. We added: *"The assumptions to the BP scheme imply that the so-called bridging stresses (also known as vertical resistive stress van der Veen and Whillans, 1989), i.e. the resistance to varying stress gradients in direction of the ice flow, is neglected. Bridging effects are generally small and occur near the ice divide and at the grounding line (Pattyn, 2000). Since the BP schemes retain stress of the order O(1) and O(epsilon) but delete stress terms of O(epsilon^2) (Blatter, 1995) it is only valid to a certain aspect ratio and topographic variability. Once high velocity gradients develop over short distances BP may not provide an accurate solution."*

Equation 9: Why breaking out 1/2 but not 1/4?
We changed the style of equation 9.

Section 2.3: Comment on that normally one would manipulate the system in the style of this page: http://websrv.cs.umt.edu/isis/index.php/Blatter-Pattyn_model   , maybe write down the "normal" BP system so that the reader more easily understands what you mean by "BP-like" in the next section
We don't think it is worth writing down the normal BP equations in this paper as they are not used/solved. We explain in detail how the classical BP equations are derived from the FS equations. We provided the Blatter and Pattyn references which show the same system manipulations as the suggested webpage.

Line 112: Add a reference for the difficulties of saddle point problems
Done.

Line 137: I worry there could still be numerical issues with how the saddle point FS system is solved
In comparison to other studies where the classical BP model is used (rearranging of equations), our approach minimizes the numerical issues by just dropping the "BP"-stress terms. The numerical problem of BP-like is therefore almost identical compared to the original FS problem. The BP-like shows almost the same convergence behaviour as FS (in terms of Newton iterations, GMRES iteration (if an iterative solver is applied), Linear and Residual error). However, we added "largely" here.

Section 3.2: Is the BP-like system symmetric? Does that matter for the linear solver you use?
No, the BP-like system matrix is still nonsymmetric. The BP-like problem poses a very similar problem as FS; the linear (and non-linear (Newton iterations)) solver converges in a very similar way for BP-like and FS.

Line 149-161: shorten this paragraph
We have shortened the description of the ASM preconditioner. However, our new simulations with the weak imposition of the friction law required a special preconditioner. The part about the solver now reads: "*For the larger problems we rely on the iterative GMRES solver (Saad, 2003) which is accelerated with appropriate preconditioners. Simulations that make use of the strong imposition rely on a Domain Decomposition solver with an overlapping additive Schwarz method (ASM, Widlund and Toselli, 2004) which is much superior in terms of the required computation time and working memory compared to MUMPS. Unfortunately, the ASM preconditioner shows a very high computational demand for simulations that employ the weak imposition. Since the involved Lagrange multiplier induces a zero on the diagonal of the system matrix, we employ the Vanka algorithm (Vanka 1986, John 2001) which is specifically designed for large indefinite problems with saddle point character. Based on our simulations, we found Vanka to be very memory efficient and computationally fast for large problems although it requires more Newton and GMRES iterations compared to ASM.*"

Line 180: The paragraph starting here can be clarified, especially for readers who does not have ISMIP-HOM details fresh in mind.
We do not understand exactly what we should explain further. In the paragraph before we provided a brief summary of the ISMIP-HOM experiments. We think that is enough without repeating the complete experimental setup.
Also I think it is the first time the abbreviation HO appears.
HO is now introduced in the Introduction

Line 193: Comment on why you choose this sliding law,
The Budd-like of friction law is often used in ice sheet modelling (Morlighem et al., 2010; Price et al., 2011; Gillet-Chaulet et al., 2012; Seroussi et al., 2013, Choi et al., 2021), it implies that the basal drag can increase without a bound. It was shown that inducing an upper ratio of $\tau_b/N$ (Iken's bound) is more justified (Iken, 1981; Schoof, 2005; Gagliardini et al., 2007; Leguy et al., 2014; Joughin et al., 2019). We choose the linear friction law (m=1) as it is commonly used in ISMs making use of inverse methods to constrain the basal friction (Morlighem et al., 2010; Larour et al., 2012; Seroussi et al., 2013; Perego et al., 2014; Gladstone et al., 2014). In the new manuscript we have now included an analysis with a power law (m=3).
and mention already here that k is to be found with inversion.
Done.

Figure 1: To me it seems the solutions in Experiment C does not seem to agree well with previous exercises. Comment on this.
We have indeed detected a minor error in the ISMIP-HOM experiments that explains the velocity amplitude disagreement. The limiter epsilon_0 in the effective strain rate was erroneously set too high. We fixed that and updated the figures. The BP-like und FS solutions agree now much better to the previous exercises (Figure R8).

[Figure]

**Figure R8:** *Results of the ISMIP-HOM experiments A (a) and C (b) for the length scale L= 5km. Surface velocity component $v_x$ at y=L/4. Values computed in this study for FS and BP-like are compared to the BP model 'rhi2' and the FS model 'oga1' from the original ISMIP-HOM benchmark (Pattyn et al., 2008). Please note that the BP-like P1P1GLS and 'rhi2' BP solutions in Exp. A overlay on each other; in Exp. C the 'rhi2' BP solution is overlaid by BP-like P2P1 strong BP-like P2P1 weak.*

Line 207-211: The boundary conditions are not completely clear to me, please write them down as equations.

We have slightly rewritten this part:

*"Solving a subset of an ice stream poses unknown boundary conditions in the interior of the ice sheet. A very simple approach would be prescribing the measured surface velocities as a depth-averaged velocity profile. However, we choose boundary conditions that are free to adjust during the solution process. Laterally boundary conditions are chosen to have a symmetry boundary condition at the inflow boundary, which implies v\*n= 0 and vanishing shear stresses. A free slip condition is chosen at lateral along-flow boundaries and a normal stress condition at the outflow boundaries. Dependent on the setup, the outflow boundaries are land- or marine terminating fronts of the glacier or located within the ice. For the former, we prescribe t·n= min(rho_i \* g (z_s−z),0), where z_s is the surface elevation and z the vertical coordinate. For the latter, we impose t\*n=rho_i\*g (z_s−z) +δ. Here, δ is an additional stress that is slightly tuned to match the observed velocities at those boundaries; it is in the order of 0.1 MPa."*

Line 215: Did you experiment with sensitivity also with respect to vertical resolution? If not please do. Note that the element aspect ratio will change if you only change horizontal resolution, probably it is not an issue here but in worst case it can impact numerics.

Now we performed a sensitivity test to the vertical resolution. See answer above (main points).

Line 222 - 225: Mention that SICOPOLIS use SIA/SSA

Done. In that case, SICOPOLIS uses SIA.

Line 226-228: This is an important point to at least discuss in your study. You find a friction coefficient that is consistent with the BP-like model, but use it also for the FS model.

We have now performed a sensitivity test on different friction fields. See answer above (main points).

Line 230-233: This paragraph seems a little bit out of place, and the table could be moved to the appendix

The Paragraph is moved. We dropped the table and provided all values in the text.

Section 5.2: Here I would appreciate a discussion relating the differences to the missing stress components, or perhaps you can just mention that it will come in section 5.4

Done.

Line 273: Mention that a discussion on why stiff ice is more sensitive will come in section 5.4

Done.

Section 5.4: I appreciate this section!

Thanks!

Line 302: Write vb/vs first, to be consistent with the order in line 300

Done.

Line 300 - 310: This is a good experiment. However the figure (Figure 7) is hard to read. Also, comment on how you think the fact the elements are flatter for high aspect ratios impact the result, or why they don't impact the result.

We added: *"Model differences emerge with increasing aspect ratio epsilon. With increasing aspect ratio deleting terms of O(epsilon^2), i.e. dv_z/dx and dv_z/dy, in the BP-like (and also BP) model becomes problematic and the solution inaccurate (Blatter, 1995)."*

Line 311-314: This is an important check. I think not all readers will understand why you look at the vertical velocity, add a sentence to explain. Perhaps even better, would be too look at how much the surface would move in one time step given this velocity field (this should be easy to compute, you don't have to actually move the surface)
See answer above (major point).

Section 5.5: I like the idea of looking at internal layers, but this part of the study is quite incomplete
We dropped this section. See answer above (major points)

Section 5.6: This section fits better after section 5.3
You are right. We moved the section accordingly.

**Minor language/esthetics comments:**

Line 26 - "Although BP neglects severe.." - is severe the right word to use?
We changed severe to several.

Line 48 - "different results as simpler models", "as" -> "compared to?"
Changed as suggested.

Line 56 - The sentence starting with "Beside the.." is a bit akward
Done. We have rewritten the sentence to: *"Utilizing FS or simpler models is not only relevant for future projections of ice sheets, the different ice dynamics may have an impact on the internal ice flow."*

Line 61 - "consistent analysis" - change for "consistent numerical experiments"?
Changed as suggested.

Line 95 - The sentence "Boundary condition ...is traction free" does not seem to be grammatically correct
We have it rewritten to *"The upper surface is assumed to be traction free."*

Line 99 - "v_b the velocity" -> "v_b is the velocity"?
Done.

Line 171: The sentence about the mmr contribution is a little bit confusing.
We deleted this sentence as it is not important for the paper.

Line 236: This sentence is hard to read
The sentence is rewritten to *"Input parameters and FS simulation results of the NEGIS subset are shown for l=6400m and 100m in Fig. 2."*

Figure 4: The scatter plots are a very small
We have dropped the scatter plots. The higher spread of E=6 compared to E=1 is already visible from the map and also shown in Fig. 5.

Line 330: This sentence is unclear, adding a "particle" or "layer" would help
Done, we added *"layers"*.

Figure 6: Increase the font
Done.

Line 395: "Seems not an urgent issue" -> "does not seem to be an urgent issue"?
Done.

------------------------- Editor ---------------------------------

I have a few more general questions and comments, which the reviewers may also touch upon. These do not have to be answered/incorporated at this stage, but I would like to invite the reviewers to consider these at a later stage of the review process, when answering the reviewer comments:

- You mention to be able to have a 'consistent' comparison, you are using an 'alternative way' to solve the BP equations. This is on the one hand interesting, as it allows making a 'cleaner' comparison of what the effect of FS vs. BP is. On the other hand, this may however raise the question how relevant it is to make such a comparison, given that others do not solve the BP system in this way (as this has a higher computational demand than the 'classic' way of solving BP, which you describe at the end of section 2.3). This does not have to be seen as something 'problematic', but it would nevertheless be good if this is brought up somewhere (e.g. in discussion), where you could also explore how different the results are between 'BP' (classic approach) and your 'BP-like' solution.
  You are right, the developed BP-like model is not intended for long time integrations or large ensembles. But we think it is a useful tool to make a clear comparison between FS and BP. The developed model is also very flexible in order to test various ice flow components (see Answer to line 367 by Reviewer #1). In our model setup, we solve for the same physical problem as in the 'classical' BP scheme but without taking advantage of the reduced number of equations to be solved. Regarding the model physics, we expect our results to compare well with other BP implementations (see the section about ISMIP-HOM, 3.3), but our BP-like model is just less performant (c.f. BP about 10 times faster than FS in Larour et al.2012). We comment on this in Sect. 3.1.

- It is really a pity that you do not have any results related to the temporal/prognostic evolution of NEGIS under BP-like vs. FS. You mention that "this is beyond the focus of this study" (l. 333) and "must be postponed to future studies" (l. 365). It is nevertheless a bit a missed opportunity to not have this here, given that you have a setup that allows for this (and which would not exclude having the diagnostic comparisons you now make). This would add some "meat to the bone" for this study. Depending on the reviewers' opinion on this, this may be an element that may need more attention in a revised version.
  See answers to the "Point (5)" at the preamble of the manuscript; additionally, our answers to the first and second major point by Reviewer #2.

- The figures nicely illustrate the results but could be improved in some cases. More specifically:
  - For some figures the labels are very hard to read (I had to strongly zoom in): please increase the fontsize in e.g. Figures 3, 4, 5 and 8
    Done. We increased the fontsize.
  - Ideally the figures can be understood as standalone figures, without having to refer to the caption (e.g. if taken as is and put in presentation). For some figures, one must carefully look at the caption to understand "what is what". Consider adding this information directly in the figure (e.g. in title): this is the case for e.g. Figures 3, 6
    The mentioned Figures 3 and 6 have the variables name at the top of the colorbar. So we think it is not necessary to add them again in the title. However, with increased fontsize it is more noticeable.

**References**

Choi, Y., Morlighem, M., Rignot, E., and Wood, M.: Ice dynamics will remain a primary driver of Greenland ice sheet mass loss over the next century, Commun. Earth Environ., 2, 26, https://doi.org/10.1038/s43247-021-00092-z, 2021.

de Boer, B., Dolan, A. M., Bernales, J., Gasson, E., Goelzer, H., Golledge, N. R., Sutter, J., Huybrechts, P., Lohmann, G., Rogozhina, I., Abe-Ouchi, A., Saito, F., and van de Wal, R. S. W.: Simulating the Antarctic ice sheet in the late-Pliocene warm period: PLISMIP-ANT, an ice-sheet model intercomparison project, The Cryosphere, 9, 881–903, https://doi.org/10.5194/tc-9-881-2015, 2015.

Durand, G., Gagliardini, O., de Fleurian, B., Zwinger, T., and Le Meur, E.: Marine ice sheet dynamics: Hysteresis and neutral equilibrium, Journal of Geophysical Research, 114, F03 009, https://doi.org/doi:10.1029/2008JF001170, 2009.

Gagliardini, O., Cohen, D., Raback, P., and Zwinger, T.: Finite-element modeling of subglacial cavities and related friction law, Journal of Geophysical Research: Earth Surface, 112, https://doi.org/10.1029/2006JF000576, 2007.

Gillet-Chaulet, F., Gagliardini, O., Seddik, H., Nodet, M., Durand, G., Ritz, C., Zwinger, T., Greve, R., and Vaughan, D. G.: Greenland ice sheet contribution to sea-level rise from a new-generation ice-sheet model, Cryosphere, 6, 1561–1576, https://doi.org/10.5194/tc-6-1561- 2012, 2012.

Gladstone, R., Schäfer, M., Zwinger, T., Gong, Y., Strozzi, T., Mottram, R., Boberg, F., and Moore, J. C.: Importance of basal processes in simulations of a surging Svalbard outlet glacier, The Cryosphere, 8, 1393–1405, https://doi.org/10.5194/tc-8-1393-2014, 2014.

Helm, V., Humbert, A. and Miller, H.: Elevation and elevation change of Greenland and Antarctica derived from CryoSat-2, The Cryosphere, 4, 1539-1559, https://doi.org//10.5194/tc-8-1539-2014, 2014

Hindmarsh, R. C. A.: A numerical comparison of approximations to the Stokes equations used in ice sheet and glacier modeling, Journal of Geophysical Research: Earth Surface, 109, https://doi.org/10.1029/2003JF000065, 2004.

Iken, A.: The Effect of the Subglacial Water Pressure on the Sliding Velocity of a Glacier in an Idealized Numerical Model, Journal of Glaciology, 27, 407–421, https://doi.org/10.3189/S0022143000011448, 1981.

John, V. and Matthies, G.: Higher-order finite element discretizations in a benchmark problem for incompressible flows, International Journal for Numerical Methods in Fluids, 37, 885–903, https://doi.org/https://doi.org/10.1002/fld.195, 2001.

Joughin, I., Smith, B. E., and Schoof, C. G.: Regularized Coulomb Friction Laws for Ice Sheet Sliding: Application to Pine Island Glacier, Antarctica, Geophysical Research Letters, 46, 4764–4771, https://doi.org/10.1029/2019GL082526, 2019.

Larour, E., Seroussi, H., Morlighem, M., and Rignot, E.: Continental scale, high order, high spatial resolution, ice sheet modeling using the Ice Sheet System Model (ISSM), Journal of Geophysical Research, 117, F01022, https://doi.org/10.1029/2011JF002140, 2012.

Leguy, G. R., Asay-Davis, X. S., and Lipscomb, W. H.: Parameterization of basal friction near grounding lines in a one-dimensional ice sheet model, The Cryosphere, 8, 1239–1259, https://doi.org/10.5194/tc-8-1239-2014, 2014.

Ma, Y., Gagliardini, O., Ritz, C., Gillet-Chaulet, F., Durand, G., and Montagnat, M.: Enhancement factors for grounded ice and ice shelves inferred from an anisotropic ice-flow model, Journal of Glaciology, 56, 805–812, https://doi.org/10.3189/002214310794457209, 2010.

Morlighem, M., Rignot, E., Seroussi, H., Larour, E., Dhia, H., and Aubry, D.: Spatial patterns of basal drag inferred using control methods from a full-Stokes and simpler models for Pine Island Glacier, West Antarctica, Geophysical Research Letters, 37, L14502, 2010.

Pattyn, F.: Ice-sheet modelling at different spatial resolutions: focus on the grounding zone, Annals of Glaciology, 31, 211–216,https://doi.org/10.3189/172756400781820435, 2000

Pattyn, F., Perichon, L., Aschwanden, A., Breuer, B., de Smedt, B., Gagliardini, O., Gudmundsson, G. H., Hindmarsh, R. C. A., Hubbard, A., Johnson, J. V., Kleiner, T., Konovalov, Y., Martin, C., Payne, A. J., Pollard, D., Price, S., Rückamp, M., Saito, F., Souˇcek, O., Sugiyama, S., and Zwinger, T.: Benchmark experiments for higher-order and full-Stokes ice sheet models (ISMIP-HOM), Cryosphere, 2, 95–108, https://doi.org/10.5194/tc-2-95-2008, 2008.

Perego, M., Price, S., and Stadler, G.: Optimal initial conditions for coupling ice sheet models to Earth system models, Journal of Geophysical Research: Earth Surface, 119, 1894–1917, https://doi.org/10.1002/2014JF003181, 2014.

Price, S. F., Payne, A. J., Howat, I. M., and Smith, B. E.: Committed sea-level rise for the next century from Greenland ice sheet dynamics during the past decade, Proceedings of the National Academy of Sciences, 108, 8978–8983, https://doi.org/10.1073/pnas.1017313108, 2011.

Saad, Y.: Iterative Methods for Sparse Linear System, Philadelphia, PA, USA: Society for Industrial and Applied Mathematics, 2003.

Schoof, C.: The effect of cavitation on glacier sliding, Proceedings of the Royal Society A, 461, 609–627, https://doi.org/10.1098/rspa.2004.1350, 2005.

Seroussi, H., Morlighem, M., Rignot, E., Khazendar, A., Larour, E., and Mouginot, J.: Dependence of century-scale projections of the Greenland ice sheet on its thermal regime, J. Glaciol., 59, 1024–1034, https://doi.org/doi:10.3189/2013JoG13J054, 2013.

Van Der Veen, C. and Whillans, I.: Force Budget: I. Theory and Numerical Methods, Journal of Glaciology, 35, 53–60,https://doi.org/10.3189/002214389793701581, 1989.

Vanka, S.: Block-implicit multigrid calculation of two-dimensional recirculating flows, Computer Methods in Applied Mechanics and Engineering, 59, 29–48, https://doi.org/https://doi.org/10.1016/0045-7825(86)90022-8, 1986.

Widlund, O. and Toselli, A.: Domain decomposition methods - algorithms and theory, vol. 34 of Springer Series in Computational Mathematics, Springer, 2005.

---

## Editor Decision (ED1)

Dear Martin Rückamp and co-authors,

Many thanks for providing this new version of your manuscript. I went through the manuscript and have provided final feedback that I would like you to consider when re-submitting your files. The list may seem somewhat long at first, but most comments should be very easy to incorporate. Once these final suggestions are addressed, we should normally be able to proceed to the acceptance of the manuscript.

- l.3: "…we explore the dynamics consequences…" → "we explore the dynamic consequences of using simplified approaches by…": i.e. suggest being a bit more specific here.
- l.9: "…whereby the BP model overestimates…"
- l.11: "…and the basal drag, where neglected stress terms in BP become important"
- l.16: "computationally intensive" → "computationally expensive"
- l.17: "…because FS effects (vs. sim models) only occur at higher resolutions"
- l.18: "…in the last decades allowed for ice flow models to directly rely on FS equations or approaches with only few simplifications"
- l.20: "require an accurate representation of stresses, e.g.…."
- l.21-22: "…necessary for long time…" (i.e. remove "e.g." here)
- l.25: "neglects several components of the full system": would be good to mention which ones here
- l.32: IPCC, 2013: true, but bit outdated. I guess a similar statement was made in SROCC or in AR6? Refer to one of these reports instead?
- l.34: "…was explored: e.g. in ISMIP-HOM…" (and remove second bracket on l.37)
- l.37: "…focused on…"
- l.38: "…show a smaller spread…" (i.e. remove, "much")
- l.38: "…and with available analytical solutions" (remove "one")
- l.39: "…was not possible…"
- l.43: "…cost when running FS models…"
- l.45-46: "…simpler models was tested on…"
- l.59: "…on FS and BP…"
- l.59: "more frequently used": compared to what? FS/SIA? Would be good to be more specific here to avoid possible confusion
- l.62:: "FS and BP intercomparison, stepping away from synthetic scenarios, and performing a high-resolution…"
- l.66: "Our study does not treat the…"
- l.68: "…12.8 km to a resolution of…" (i.e. remove "down")
- l.71: given the importance of the full-Stokes (FS) equations in your storyline, would it make sense to also explain shortly how this relates to navier-Stokes, which may be more familiar to those outside the glaciological community?
- l.104: not sure if can have colours in equations in final document. Needs to be seen with copy-editing. Suggest adding a note to flag this.
- l.117-118: "…van der Veen and Whillans"
- l.122: "reduces the computational demand": could you give an indication by how much this is the case?
- l.126: "…alternative way to align with the BP stress regime, thereby directly allowing to compare FS and HO model simulations": the second part of the sentence that I suggest to add is not a must, but think it would be nice to explicitly mention this here to conclude this section.
- l.130: remove bracket after "2"
- l.132: "…in order to align with the BP stress regime"
- l.133: "That means that" → "Through this approach,…"
- l.130-140: nicely formulated. Very clear, also for non-expert!
- l.159: "increases" → "increase"

- l.163: "…which requires more computation time and working…" (or less computation, in case I did not understand this correctly. There's some room for misinterpretation in actual formulation, therefore good to change also)
- l.183: "Exp. C is a parallel-sided slab"
- l.187: "In the parallel direction, 15 layers…": in general, many instances throughout the text where a "," could/should be added. Could check for this, although will normally be done through copy-editing phase.
- l.193-194: "reveals distinct differences between FS and lower order approximations": not entirely clear here: is this FS vs. HO? Or vs. SIA? Would be good to be more specific here.
- l.195: "On those": shorter time scales? Would be good to specify what those refers to here
- l.197: "… we found a good agreement with the original ISMIP-HOM contributions and within our own model versions": not sure if with the latter 'within' the sentence expresses what you wanted to say. But thought this'd be clearer than original formulation. If disagree, feel free to leave as is of course
- l.203: "…focuses on…"
- l.204: "The second region also includes the…"
- l.207-208: "bed and surface topography": what is source for surface topography in BedMachine?
- caption figure 1: "overlay on each other" → "overlap"?
- l.214-15: when referring to the 'shock': suggest to also directly refer to the ISMIP initialization experiments (Goelzer et al., 2018, TC) here, as they clearly make this point
- l.218: "…allow for a better comparison"
- l.220: "…is assumed to be:"
- l.229: density of the ocean water. Possibly add a reference for taking this value?
- l.236: "slip-free condition"?
- l.238: "For those outflow boundaries…"? Not sure, but would be good to specify what the "those" refers to here
- l.242: "…ZI; where…"
- l.253: "…from a few seconds to several hours": also the upper limit does not seem to be a lot. How does this align with the statement made in the previous line which refers to this being expensive?
- l.260: "is computed by Cuffey and Paterson" → "is computed following Cuffey and Paterson" + also further in sentence when referring to Lliboutry and Duval (1985). Otherwise sounds like Cuffey and Paterson calculated this themselves... For the reference to the value by Cuffey and Paterson, suggest to also include the Table where this comes from in the book (if I'm not mistaken there's several values suggested there, no?)
- Table 1:
    - "…MPI)) are kept constant between…"
    - "…hybrid mode, meaning the process…"
    - "Each production node has 362 GB RAM and…"
- l.268: "ice-stream region for l=6400 m and 100 m is provided in Fig. 3"
- l.269-270: "…(Fig. 3b,e) at higher resolution reveals… a rapidly varying bed" (remove "in the finer resolution" at end of the century)
- l.278: "…could certainly be…"
- l.285: "The largest impact of using a BP-like solution results from the surface flow velocities, which are up to 43 ma$^{-1}$ faster (…"
- l.290: "…spread in differences is more pronounced for…"
- Fig. 5 caption: "…for E = 0.1,…": remove "or"
- l.311-312: split in two sentences: "(Fig. 6c,f). However,…"
- l.316: "difference is" or "differences are"
- Fig. 6 caption: not easy to follow. Suggest putting the references to the panels before the explanation: i.e. "(a) slope of bed topography, (b) basal friction coefficient for E=1 and (c) simulated surface…

- Fig. 8 caption: "For those": which ones? Please specify
- l.332: "Overall, the BP-like model results in higher discharge than FS, while the opposite is true for 79NG and ZI."
- l.344: "Due to the hyperbolic…"
- l.345: "…which smooths the sharp gradients…"
- l.347: "Recalling values": what are these? Not entirely clear. Possibly reformulate?
- l.352: suggest removing "certainly"
- l.359: "hinge zone": what is this? Would be good if you could explain this here
- l.361: "…complex composition of ice dynamic…"
- l.364: "…flavour of the origins…": not sure what you are referring to here. Could you specify?
- Figure 10: "Lower shows…" → "Panels (c) and (d) show the…"
- l.365: "Figure 11 and 12 shows" → "Figure 11 and 12 show"
- l.370: "…lower in some larger areas"
- l.374: "…in the majority larger…" → "…is generally larger for…"
- l.379: "weak" and "strong" setup: is not entirely clear what this is here.
- l.386: "…, in the BP-like…" → ", the BP-like"
- l.387-388: "Similarly, the difference between FS and BP is most pronounced in region with a high slip ratio…, while with a higher resolution…": not sure about the second part ("while with…"), as the sentence was not entirely clear to me. If not correctly reformulated, try to opt for slightly different formulation to better reflect what you mean here.
- l.391: "0 to 0.05, the higher bed undulations lead to larger errors at higher resolutions; however it…": not entirely clear to what the last "it" refers here: (the neglection of) the bed slope?
- l.393: would be good to also briefly refer to work on (mountain) glaciers that has shown this also. Particularly thinking of Le Meur et al. (2004, https://doi.org/10.1016/j.crhy.2004.10.001) for this.
- l.397: "are to date rarely used…": would be good to specify this, as will change in the (near) future
- l.398: "we found distinct differences" + l.399: "these differences": not entirely clear, would be good if you could specify differences in what?
- l.400: "…that at a higher resolution the disagreements would be more substantial, but…"
- l.400: bedrock topography database: how is this linked to the disagreements you mention before? Not entirely clear. Suggest reformulating.
- l.401: "…computationally expensive, it…"
- l.402: "…that are currently being run within the ice sheet modelling community": see comment above also
- Figure 11 caption:
  - "Pas" → "Pa s"?
  - "that range from E=1 to E=6"
- Figure 12 caption: "…equivalent fields but for m=3", correct?
- l.409-410: "Figure 15 (panels a and c) shows the…"
- l.414: "…that is for most areas higher…"
- l.416: "…made to BP…": not entirely clear here. Maybe simply omit?
- l.432: "…using the geometry from…"
- l.434: "…respond somewhat similarly (see…"
- l.435: "…coverage or the fact that ZI has…"
- l.438: "…and analysing how the…"
- l.467: "…must be investigated in future studies"
- l.468: "…within a thermo-mechanical model?"
- l.469: "…ice flow. Additionally, there…": as is not really in contrast with what you have said before it seems. Seems more like an extra suggestion.
- l.480-481: suggest omitting "An intriguing effect was identified" at start of the sentence and also reformulate in general: "In the ice stream region, when considering a rheology with much

softer ice (E=6) than typically considered, and enhanced spread…and BP-like was modelled":
than typically considered → why is this?

- l.481: "In contrast, stiffer ice (E=0.1) leads to…": was not clear what the reference is for 10 time stiffer. Therefore suggested removing this.
- l.484: "…as important surface velocity differences occur (up to …of 79NG). Numerical…"
- l.485-486: "…to the choice of friction type: model differences are particularly important when a power-law friction is applied (as opposed to a linear approach)"
- l. 491: "…results from non-FS models…": otherwise a huge range of models is possible (e.g. SIA, SIA-SSA,..etc)
- l.492: "…software that is not freely…"
- l.494: remove "(" before "Larour"
- Acknowledgments: please acknowledge the input from the reviewers

Best regards,
Harry

---

## Author Response (AR2)

We would like to thank the reviewers for their constructive comments that helped to improve the manuscript. We have revised the manuscript accordingly and will be happy to provide a new manuscript. Please find below the reviewer's comments in black and a point-by-point response in blue.

The authors have significantly improved the manuscript, addressing my concerns and those of the other reviewer. They have broadened the scope by adding an analysis of the NEGIS outlet region and by testing a basal power law and various numerical differences, while removing the advection analysis that seemed too limited. They have also clarified some technical and conceptual details and modified the conclusions appropriately.

I have a couple of substantive comments, followed by some suggested minor corrections. Here and below, page and line references are to the version with tracked changes.

The first comment is related to the new Section 5.4. The vertical emergence velocities are large, ~50 m/a. This number is described (l. 390) as "not very extreme", but it exceeds observed height changes by a factor of ~25. What explains these large values? Is there a mismatch between the computed velocities and the observationally derived basal topography? If so, then the emergence velocity is basically a measure of the mismatch. With either stress balance, the model would presumably relax in a few years to a smooth surface slope with a modified velocity field and much smaller emergent velocities.

The large values could be partly explained by the high resolution (up to 100m) employed in this study. At the region where the high emergence velocities develop, the surface slopes ($\sqrt{n_x^2+n_y^2}$) are exceeding values of 0.1. Assuming a surface velocity of about 1000m/a, the emergence velocity is about 100m/a. As also mentioned in the text (line 390-392), the high values could also be computed ($v_x*n_x+v_y*n_y$) when using the observationally derived products, i.e. the GIMP height elevations and the MEASURE velocities.

Certainly, the emergence velocity will reduce when running a transient relaxation run as the setting will reach a steady-state. However, we do not think that the high values are critical, as we are interested in the differences between FS and BP-like.

We added here: *"A transient relaxation run would certainly smooth the surface slopes. However, the high values are not critical as we are interested in the differences between FS and BP-like."*

The details of the relaxation would differ between FS and BP, but I'm not clear on why these details are important for ice sheet models. In particular, I'm not sure how we know that the relaxed surface would be lower for BP than for FS (l. 396).

The surface for BP is expected to be lower as the emergence velocity is lower (see red patches in Figs. 10b and d). This very pronounced in the grounding zone of 79NG. Since the emergence velocity of BP is lower than in FS, it would predict a lower surface than FS within the first time step (assuming SMB=0 or the same SMB in both models). This finding is consistent with the higher discharge in BP and a grounding line position that is located further upstream compared to FS. We have rewritten the sentence in Line 396: *"The emergence velocity differences between FS and BP-like reveals a higher emergence velocity in FS, particularly at the grounding line at 79NG. According to Eq. 14, BP-like would therefore compute a surface elevation that is lower than the FS surface within the first time step."*

Without a better explanation of why the emergent velocity matters, I suggest removing this section.

We think the evaluation of the emergence velocity is an important diagnostic in order to estimate how the model differences could evolve over time. Additionally, the second reviewer recommended this analysis to overcome the lack of time dependence in our study.

Second, the authors now claim (p. 26, l. 469) that based on the new analysis of NEGIS outline regions, "ice flow treatment with FS is essential". This is a strong claim, suggesting that models with BP (or L1L2 or SSA) solvers should switch to FS, regardless of the increased complexity and computational cost. Indeed, the new analysis shows that FS–BP differences are large in the 79N grounding zone. But modelers could compensate for these differences (e.g., by tuning basal drag coefficients), and the differences might matter more or less depending on the science application. I think it would be enough here (as in the Abstract, where the wording is more measured) to say that the BP– FS differences are large, and we should take these differences into account when evaluating results from models with BP or simpler schemes.

We agree with the reviewer that this is a strong claim and needs further research to prove (e.g., transient *simulations). Therefore, we have rewritten the sentence following the reviewer's suggestion: "However, once the outlet regions of NEGIS are included (i.e., 79NG and ZI), the analysis shows that differences between FS and BP-like are large in the 79N grounding zone."*

I'd make a similar comment about the last sentence of the Conclusions (p. 31, l. 610): "a correct representation of the ice dynamics in critical areas is required." Of course, one wants an accurate representation in critical areas, but no model is perfectly correct. It might be more constructive to say that there are regions where FS and BP differ significantly, and where results from simpler models should therefore be viewed with caution.

Done. We followed the reviewer's suggestion hand have rewritten the sentence to: *"Our diagnostic simulations and previous studies indicate that FS and BP differ significantly in regions with a grounding zone and results from simpler models should therefore be viewed with caution."*

Minor corrections:

• P. 1, l. 14: "unveil" -> "show". In general, don't use "unveil" as a synonym for "show".
Done
• P. 3, l. 77: Here and below, I suggest "regions" (a geographic term) instead of "subsets" (a mathematical term).
Done.
• P. 4, l. 100: Delete "is defined"
Done
• P. 4, l. 112: "For floating ice"
Done
• P. 4, l. 113: Delete "used"
Done
• P. 5, l. 127: Delete "up-to-date"
Done
• P. 5, l. 136: Delete "the order"
Done.
• P. 6, l. 164: "circumvented"

Done.

• P. 7, l. 178: Update the last-access date if possible

Done.

• P. 7, l. 191: Add comma after "matrix". This is one of several places where adding a comma after an initial clause or prepositional phrase would help the reader.

Done.

• P. 7, l. 199: "Dependent" -> "Depending", comma after "discretization, "run" -> "runs

Done.

• P. 7, l. 200: Not clear what is meant by "The limiting case MUMPS and ASM"

We have rewritten this sentence to: *"For the largest DOF model that could be solved with the direct solver on our cluster system, we also performed a simulation with the iterative linear solver."*

• P. 7, l. 206: Use quotes for case names consistently (either with or without)

Done. We added quotes to *Ice flow over a bumpy bed*.

• P. 8, l. 211: Add comma after "margins"

Done

• P. 8, l. 218: "using" -> "use"

Done.

• P. 8, l. 234: "upstream of", "downstream of".

Done.

• P. 9, l. 242: I'm not clear on what's meant by "A relaxation run based on e.g., the BP-like scheme and successive simulation runs with BP-like and FS"

We replaced "successive" with "subsequent".

• P. 10, Fig. 2 caption: The two domains (red and cyan) partly overlap. Here or in the main text, could you explain why this is the case?

There is no specific reason why both regions partly overlap. Both regions were designed to capture slow and fast flowing regions of the NEGIS.

• P. 11, l. 277: "used contained" is awkward. Maybe "span a typical range of resolutions…"

Done

• P. 11, l. 284: Delete "very"

Done.

• P. 11, l. 288: "to" -> "too"

Done.

• P. 12, Table 1: Has formatting issues in this version, but looks OK in the other version

Ok, we will take care of this in the revised version.

• P. 15, Fig. 4: The figure show v_FS – v_BP. Thus where BP is faster, Fig. 4 shows negative values, plotted in blue. This is a bit counterintuitive. If FS is "truth", then it's more natural, at least for me, to show the BP error with respect to truth, i.e. red where BP is too fast. If it's not hard to change the figures, I suggest flipping the sign.

We agree with the reviewer that another choice on how to calculate the differences would be more intuitive (i.e., BP-FS -> BP faster than FS -> positive sign -> 'red color'). However, to our knowledge there is no rule on how to calculate differences of data (i.e., BP) to 'truth'/observation (i.e., FS). The calculation of the differences is clearly mentioned in the text and figure captions so there should be no confusion. However, generally we would follow this suggestion but it requires a lot of figure changes (Figs. 4, 5, 7, and 8) and corresponding text changes (flipping sign of mentioned values). So, we do not follow this suggestion.

• Sections 5.2 and 5.3: The new analyses in these sections definitely strengthen the paper.

Thanks. The paper benefits from your suggestions from of the first review.

• P. 16, l. 351: Delete "Equivalent" and "exemplary for". The word "exemplary" is misused a couple of times; could also delete in the Fig. 6 caption.

Done.

• P. 17, l. 355: "expressed" -> "pronounced"

Done.

• P. 17, l. 359: "stronger" -> "larger"

Done.

• P. 17, l. 367: "Most remarkably" implies that this is a surprising result, but I think it's to be expected. Maybe "notably" instead.

Done.

• P. 18, Fig. 7: Same comment as for Fig. 4. It would be more intuitive to use red for regions where BP is too fast.

see above

• P. 18, l. 374: The wording here makes it unclear what the numbers (14.9 and 14.2) refer to. Are these the fluxes for 79NG and ZI, respectively, when simulated by FS? Similarly in the Fig. 9 caption. Clearer wording would be "On average, the total simulated flux [for FS?] is 14.7 Gt a^-1 for 79NG and 14.2 Gt a^-1 for ZI."

You are right the wording is unclear. Yes, it is the average over all settings (resolutions, discretizations, BC implementations, friction exponents) when simulated with FS.

• P. 20, Fig. 9: Similar comment to Figs. 4 and 7. Since BP overestimates fluxes compared to the more correct FS, it would be natural to show the relative Q difference as positive.

see above

• P. 22, l. 425: "stronger pronounced" -> "more pronounced"

Done

• P. 22, l. 434: Add comma after "ratio"

Done

• P. 25, l. 458: Delete "with each other"

Done

• P. 26, l. 464: Delete the second "much"

Done.

• P. 29, l. 565: "less performant" -> "more expensive"

Done.

• P. 30, l. 592: Not sure what "offset" means here.

We have rewritten the sentence to: "In contrast, ten times stiffer ($E=0.1$) ice leads to enhanced ice flow velocities in BP-like compared to FS.".

• P. 30, l. 595: See the comment above on "essential"

Done. See answer above.